EMBO
Molecular Medicine

# Cholesterol metabolism promotes B-cell positioning during immune pathogenesis of chronic obstructive pulmonary disease

Jie Jia[1,2,†], Thomas M Conlon[1,2,†] (iD), Rim SJ Sarker[1,2], Demet Taşdemir[3], Natalia F Smirnova[1,2],
Barkha Srivastava[1,2], Stijn E Verleden[4], Gizem Güneş[1,2], Xiao Wu[5], Cornelia Prehn[6,7], Jiaqi Gao[1,2],
Katharina Heinzelmann[1,2], Jutta Lintelmann[5], Martin Irmler[8], Stefan Pfeiffer[9], Michael Schloter[9],
Ralf Zimmermann[5,10], Martin Hrabé de Angelis[7,8,11], Johannes Beckers[7,8,11], Jerzy Adamski[6,7,11],
Hasan Bayram[3,12], Oliver Eickelberg[1,2,13,*] (iD) & Ali Önder Yildirim[1,2,**] (iD)

## Abstract

The development of chronic obstructive pulmonary disease (COPD) pathogenesis remains unclear, but emerging evidence supports a crucial role for inducible bronchus-associated lymphoid tissue (iBALT) in disease progression. Mechanisms underlying iBALT generation, particularly during chronic CS exposure, remain to be defined. Oxysterol metabolism of cholesterol is crucial to immune cell localization in secondary lymphoid tissue. Here, we demonstrate that oxysterols also critically regulate iBALT generation and the immune pathogenesis of COPD. In both COPD patients and cigarette smoke (CS)-exposed mice, we identified significantly upregulated CH25H and CYP7B1 expression in airway epithelial cells, regulating CS-induced B-cell migration and iBALT formation. Mice deficient in CH25H or the oxysterol receptor EBI2 exhibited decreased iBALT and subsequent CS-induced emphysema. Further, inhibition of the oxysterol pathway using clotrimazole resolved iBALT formation and attenuated CS-induced emphysema in vivo therapeutically. Collectively, our studies are the first to mechanistically interrogate oxysterol-dependent iBALT formation in the pathogenesis of COPD, and identify a novel therapeutic target for the treatment of COPD and potentially other diseases driven by the generation of tertiary lymphoid organs.

**Keywords** B cell; chronic obstructive pulmonary disease; inducible bronchus-associated lymphoid tissue; oxysterol; tertiary lymphoid organ
**Subject Categories** Immunology; Respiratory System

## Introduction

Chronic obstructive pulmonary disease (COPD) is a leading cause of chronic mortality and morbidity worldwide with limited therapeutic options, characterized by progressive and largely irreversible airflow limitation resulting from long-term exposure to toxic gases and particles, in particular cigarette smoke (CS; Berndt *et al*, 2012; Vogelmeier *et al*, 2017). This induces chronic bronchitis, small airway remodeling, and emphysema (loss of septal tissue; Tuder & Petrache, 2012). Growing evidence supports a crucial role for inducible bronchus-associated lymphoid tissue (iBALT) in the development of COPD (Hogg *et al*, 2004; Gosman *et al*, 2006; van der Strate *et al*, 2006; Polverino *et al*, 2015; Faner *et al*, 2016). Furthermore, we and others have recently shown that an absence of

1 Comprehensive Pneumology Center (CPC), Institute of Lung Biology and Disease, Helmholtz Zentrum München, Munich, Germany
2 Member of the German Center for Lung Research (DZL), Munich, Germany
3 Department of Chest Diseases, School of Medicine, University of Gaziantep, Gaziantep, Turkey
4 Division of Pneumology, KU Leuven, Leuven, Belgium
5 Joint Mass Spectrometry Centre, Comprehensive Molecular Analytics, Helmholtz Zentrum München, Munich, Germany
6 Institute of Experimental Genetics, Genome Analysis Center, Helmholtz Zentrum München, Munich, Germany
7 German Center for Diabetes Research (DZD), Munich, Germany
8 Institute of Experimental Genetics, Helmholtz Zentrum München, Munich, Germany
9 Research Unit Comparative Microbiome Analysis, Helmholtz Zentrum München, Munich, Germany
10 University of Rostock, Rostock, Germany
11 Chair of Experimental Genetics, Technische Universität München, Freising-Weihenstephan, Germany
12 School of Medicine, Koç University, Istanbul, Turkey
13 Division of Pulmonary Sciences and Critical Care Medicine, University of Colorado, Denver, CO, USA
*Corresponding author. Tel: +1 303 724 4075; E-mail: oliver.eickelberg@ucdenver.edu
**Corresponding author. Tel: +49 89 3187 4037; E-mail: oender.yildirim@helmholtz-muenchen.de
†These authors contributed equally to this work

iBALT, either through the use of B cell-deficient mice or administration of anti-CXCL13 antibody or BAFF-receptor fusion protein, prevented CS-induced emphysema in animal models of COPD (Bracke *et al*, 2013; John-Schuster *et al*, 2014; Seys *et al*, 2015).

Inducible bronchus-associated lymphoid tissue is a tertiary lymphoid organ that develops in the lungs during infection, autoimmunity, or chronic inflammation (Rangel-Moreno *et al*, 2011; Hwang *et al*, 2016). Interestingly, iBALT is located predominantly alongside the bronchial epithelium (Gregson *et al*, 1979). It is organized into regions of B-cell follicles surrounded by T-cell zones reminiscent of conventional secondary lymphoid organs (Randall, 2010). Follicular dendritic cells and high endothelial venules are located in the B- and T-cell zones, respectively (Moyron-Quiroz *et al*, 2004; Rangel-Moreno *et al*, 2007). A protective role for iBALT has been described due to their ability to fight viral infection (Moyron-Quiroz *et al*, 2004; Chiu & Openshaw, 2015), and however, they can have a detrimental impact on the outcome of chronic inflammatory conditions such as COPD (Hwang *et al*, 2016). Their causative role against COPD development and the mechanism underlying iBALT positioning upon the bronchus, however, remains to be defined.

The oxysterol metabolism of cholesterol has recently emerged as a central pathway that regulates the structure and function of secondary lymphoid tissue (Hannedouche *et al*, 2011; Liu *et al*, 2011; Gatto *et al*, 2013; Li *et al*, 2016). The sequential action of two enzymes cholesterol 25-hydroxylase (CH25H) and 25-hydroxy-cholesterol 7-alpha-hydroxylase (CYP7B1) synthesizes 7α,25-dihydroxycholesterol (7α,25-OHC) from cholesterol, the main ligand of Epstein–Barr virus-induced gene 2 (EBI2; also known as GPR183; Hannedouche *et al*, 2011; Liu *et al*, 2011). EBI2 is a G protein-coupled receptor expressed on lymphocytes and dendritic cells (DCs), which plays a key role in their positioning within secondary lymphoid tissue. EBI2 on B cells (Gatto *et al*, 2009; Pereira *et al*, 2009) and 7α,25-OHC generated by lymphoid stromal cells (Yi *et al*, 2012) guides activated B-cell movement during humoral responses.

In this study, we hypothesize that oxysterols are critically involved in the immune pathogenesis of COPD by guiding B-cell positioning within iBALT structures. We demonstrate, in both COPD patients and CS-exposed mice, upregulated CH25H and CYP7B1 expression in airway epithelial cells. Moreover, mice deficient in CH25H or EBI2 exhibited decreased iBALT formation and subsequent CS-induced emphysema. Further, activated B cells through BCR cross-linking failed to migrate toward CS-stimulated airways *ex vivo* following genetic or pharmacological inhibition of the oxysterol pathway, establishing a role for oxysterol metabolism in guiding iBALT generation to the airways during COPD immunopathogenesis. Finally, inhibition of the oxysterol pathway, using the CYP7B1 inhibitor clotrimazole, resolved B cell-driven iBALT formation and attenuated CS-induced emphysema *in vivo* in a therapeutic approach. Collectively, our studies are the first to mechanistically interrogate oxysterol-dependent iBALT formation in the pathogenesis of COPD, and identify a novel therapeutic target for the treatment of COPD in particular, as well as other chronic diseases driven by the generation of tertiary lymphoid organs.

# Results

## Oxysterol metabolism increases in airway epithelial cells of COPD patients and mouse

Airway epithelial cells secrete a plethora of immune mediators (Benam *et al*, 2016), yet immunological factors that orchestrate iBALT positioning remain to be defined. We analyzed transcriptomics data from publically available datasets of small airway epithelial cells from COPD patients (Tilley *et al*, 2011) combined with our data derived from chronic CS-exposed mice lungs (John-Schuster *et al*, 2014), revealing a conserved interspecies signature for the expression of key genes related to the Gene Ontology terms (Wang *et al*, 2013) of "Inflammatory Response", "Macrophage Activation", and "Leukocyte Migration" (Fig EV1A). Gene expression patterns were also similar within the Gene Ontology term "Metabolic Process", and in particular, *CH25H* and *CYP7B1* were upregulated following both CS exposure in mice and in COPD patients (Fig 1A). Similarly, RNAseq analysis of lung homogenates from an independent COPD patient cohort confirmed higher *CH25H* expression in the lungs of COPD patients compared to non-smoking control individuals (Fig 1B), supporting a previous study (Sugiura *et al*, 2012). To demonstrate an association with emphysema, lung core samples from a third independent cohort composed of emphysematous and non-emphysematous tissue from the same COPD patients were analyzed (Fig 1C). mRNA expression of *CH25H* and the pro-inflammatory chemokine *CXCL8* were significantly upregulated in emphysematous regions rather than non-emphysematous regions of COPD patient lungs, while in contrast to recent findings (Faner *et al*, 2016), *CXCL13* expression did not differ (Fig 1D). Staining of airway sections revealed that CH25H was localized to the airway epithelial cells in both human and mice (Fig 1E), suggesting that the initiating lesion in both patients and mice following chronic CS exposure emanates from the airways. *CH25H* mRNA expression was elevated in isolated airway epithelial cells from COPD patients compared to healthy smoking controls (fourth independent cohort; Fig 1F), as well as in isolated mouse airways after CS exposure, and remained elevated for at least 16 weeks (Fig 1G). Bronchoalveolar lavage fluid obtained from mice exposed to 6 months chronic CS revealed a higher concentration of 25-hydroxycholesterol as assessed by liquid chromatography–high-resolution mass spectrometry (Fig 1H).

To address the mechanism underlying the regional upregulation of CH25H predominantly localized to the airways in COPD patients and in particular that associated with emphysematous tissue, we first undertook gene set enrichment analysis (GSEA; Mootha *et al*, 2003; Subramanian *et al*, 2005) on the publically available transcriptomics dataset of small airway epithelial cells from COPD patients described above (Tilley *et al*, 2011). CH25H upregulation is driven by TLR4 Myd88-independent signaling (Diczfalusy *et al*, 2009), and indeed, GSEA revealed a strong enrichment of both total TLR- and TLR4-dependent signaling in small airway epithelial cells taken from the lungs of COPD patients compared to smoking controls (Fig EV1B). Supporting a recent observation that *TLR4* expression is increased in the airways of COPD patients (Haw *et al*, 2017). Furthermore, staining of airway sections revealed a strong increase in TLR4 expression localized to the airways of emphysematous COPD patients rather than non-emphysematous or healthy

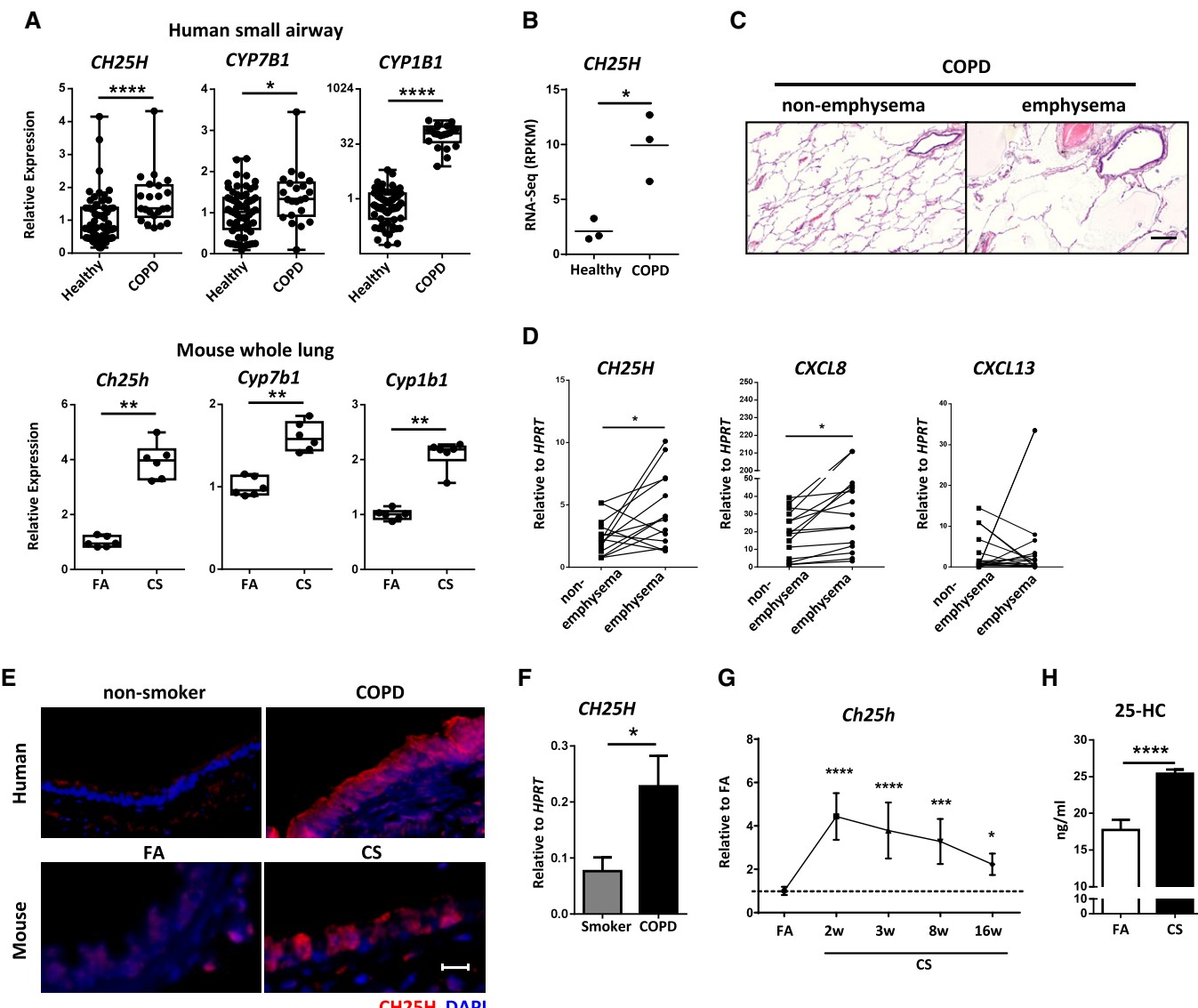

**Figure 1.  Increased expression of the oxysterol metabolizing enzyme CH25H in airway epithelial cells of COPD patients.**

A   Box and whisker plots of mouse lung and human small airway epithelial cell microarray data of the relative expression of the genes from NCBI GEO data series. The box extends from the 25th to 75th percentiles with the central line indicating the median and the whiskers the minimum and maximum values. Each dot represents an individual. FA, filtered air; CS, cigarette smoke. *P = 0.0119, **P = 0.0022 (*Ch25h, Cyp7b1,* and *Cyp1b1*) and ****P < 0.0001.

B   RNAseq data of *CH25H* expression in an independent COPD cohort, three patients per group. *P = 0.0136.

C   Representative H&E-stained lung cores from non-emphysematous and emphysematous regions of the same COPD patient lung (fourteen patients). Scale bar, 500 μm.

D   *CH25H, CXCL8,* and *CXCL13* mRNA abundance from lung core samples described in (C). Individual patients shown. *P = 0.0261 (*CH25H*) and 0.0482 (*CXCL8*).

E   Representative immunofluorescence analysis of airway from lungs of human non-smokers or COPD patients and from mice exposed to filtered air (FA) or cigarette smoke (CS) for 6 months, stained to detect CH25H (red) and DAPI (blue). Four to eight patients per group, and eight mice per group. Scale bar, 50 μm.

F   *CH25H* mRNA abundance in isolated airway epithelial cells from smokers (n = 10) and COPD patients (n = 11). *P = 0.0236.

G   *Ch25h* mRNA abundance in isolated airways from C57BL/6 mice exposed to cigarette smoke (CS) for the duration indicated, shown relative to filtered air (FA), one experiment with five mice per group. *P = 0.0377, ***P = 0.0002, and ****P < 0.0001.

H   Levels of 25-hydroxycholesterol in the bronchoalveolar lavage fluid of C57BL/6 mice exposed to FA or CS for 6 months as determined by liquid chromatography–high-resolution TOF mass spectrometry, one experiment with four mice per group. ****P < 0.0001.

Data information: Data are mean ± SD (F, G, H). Mann–Whitney test (A, F), two-tailed unpaired *t*-test (B, H), two-tailed paired *t*-test (D) or one-way ANOVA, and uncorrected Fisher's LSD (G).

control airways (Fig EV1C). Additionally, treating human bronchial epithelial cell lines with the TLR4 agonist LPS induced expression of *CH25H* similar to that observed with cigarette smoke (Fig EV1D and

E). Interestingly, the pro-inflammatory cytokine TNF-α alone was also able to induce enhanced *CH25H* expression in airway epithelial cells, suggesting that the pro-inflammatory environment in addition

to direct effects of CS exposure upon the airway epithelial cells is capable of enhancing *CH25H* expression. These translational results lead us to hypothesize that CS-activated CH25H signaling in the airway epithelium may confer iBALT formation.

## Diminished oxysterol pathways impaired iBALT formation and attenuated cigarette smoke-induced COPD

To determine the role of CH25H in iBALT formation *in vivo*, we exposed CH25H-deficient mice to CS for 4 and 6 months to induce emphysema development (John-Schuster *et al*, 2014; Cloonan *et al*, 2016; Baarsma *et al*, 2017). Importantly, the lungs of these mice showed no general metabolic differences, even after CS exposure, compared to the wild-type animals (Fig EV2A). Wild-type mice developed clear evidence of emphysema accompanied by iBALT formation from 4 months onwards (Fig 2A–C), that specifically associated with the airways further with vessels and septal tissue (Fig 2D), whereas in *Ch25h*$^{-/-}$ mice formation of iBALT and the hallmarks of emphysema failed to develop (Fig 2A–D). Flow cytometric analysis of whole lung cells showed that both T and B cells were activated after CS exposure similarly in both *Ch25h*$^{-/-}$ and wild-type mice (Fig 2E), suggesting that CH25H is important for cellular positioning within the iBALT and not recruitment and activation of T and B cells to the lung. In support, cellular recruitment into the BAL as well as mRNA expression of *Cxcl13*, *Cxcl9*, *Ccl19*, *Ccl21*, *Cxcl1*, and *Mcp1* was equivalently increased in both wild-type and *Ch25h*$^{-/-}$ mice following CS exposure (Fig EV2B and C).

We have previously shown that B cell-deficient mice do not generate iBALT and that this prevented CS-induced emphysema by impairing the activation of macrophages and MMP12 upregulation (John-Schuster *et al*, 2014). To address the mechanisms underlying the protection against emphysema in *Ch25h*$^{-/-}$ mice, flow cytometric analysis was undertaken on the lavaged lungs to address the recruitment of macrophages into the lung tissue following CS exposure. In contrast to the BAL, total F4/80$^+$ macrophages and CD11c$^-$CD11b$^+$ recruited macrophages were significantly reduced in the lungs of *Ch25h*$^{-/-}$ mice compared to wild-type animals following chronic CS exposure (Fig EV2D–F). In support, immunohistochemically stained galectin-3-positive macrophages (Fig EV2G), mRNA expression of *Adgre1* the gene for F4/80 (Fig EV2H), and the *Mmp12:Timp1* ratio (Fig EV2I) were significantly reduced in the lungs of *Ch25h*$^{-/-}$ mice compared to wild type, following exposure

to CS. Furthermore, flow cytometric analysis revealed reduced Ly6g-positive neutrophils in the lungs of *Ch25h*$^{-/-}$ mice compared to wild-type animals following chronic CS exposure (Fig EV2J–K).

To confirm the importance of local oxysterol production in iBALT generation and COPD pathogenesis, we next exposed mice lacking EBI2, the receptor for 7α,25-OHC, to chronic cigarette smoke for 4 months. Similar to *Ch25h*$^{-/-}$ mice, these *Ebi2*$^{-/-}$ animals also failed to generate iBALT or any features of emphysema (Fig 3A–C). Leukocyte recruitment to the bronchoalveolar lavage fluid following CS exposure was similar in both wild-type and *Ebi2*$^{-/-}$ mice (Fig EV3A), with cytokine and chemokine expression profiles similar to that observed for *Ch25h*$^{-/-}$ and wild-type mice (Fig EV3B). Flow cytometric analysis of whole lung cells revealed that both T and B cells were recruited to the lungs of *Ebi2*$^{-/-}$ mice in similar numbers to wild-type animals following CS exposure (Fig 3D and E), but interestingly less B cells were activated in the *Ebi2*$^{-/-}$ mice (Fig 3D), suggesting a further role for EBI2 beyond B-cell positioning (Benned-Jensen *et al*, 2011). To address this, splenic B cells were isolated from *Ebi2*$^{-/-}$ and wild-type mice and activated *ex vivo* by IgM cross-linking. Similar to the *in vivo* situation flow cytometric analysis revealed reduced activation of *Ebi2*$^{-/-}$ B cells as demonstrated by less upregulation of the surface activation marker CD69 (Fig EV3C), which was accompanied by reduced MHC II expression (Fig EV3D). CD69 expression in B cells is regulated by Egr1 (Richards *et al*, 2001; Vazquez *et al*, 2009), a primary response gene rapidly induced in B cells following BCR cross-linking (Seyfert *et al*, 1989; McMahon & Monroe, 1995). Interestingly, *Ebi2*$^{-/-}$ B cells 6 h post-BCR cross-linking upregulated *Egr1* less than wild-type B cells (Fig EV3E), proposing that the impaired activation of *Ebi2*$^{-/-}$ B cells may stem from an inability to fully induce expression of the early response gene *Egr1*. Future work should determine further the role of Ebi2 in *Egr1* transcriptional regulation.

To exclude a role for 25-hydroxycholesterol in emphysema development beyond iBALT formation, we examined B cell-deficient mice (μMT) that are resistant to CS-induced iBALT formation and emphysema (John-Schuster *et al*, 2014). Consistent with wild type, these animals demonstrated increased expression of *Ch25h* and *Cyp7b1* (Fig EV4A and B) in their lungs with CH25H expression localizing to the airway epithelial cells (Fig EV4C) following acute and chronic CS exposure. Collectively, these data demonstrated that oxysterol-induced signaling pathways guide iBALT generation during CS-induced COPD.

---

**Figure 2. Impaired iBALT formation and attenuated cigarette smoke-induced COPD in CH25H-deficient mice.**

A    Representative H&E-stained lung from wild-type (WT) and CH25H-deficient (*Ch25h*$^{-/-}$) mice exposed to filtered air (FA) or cigarette smoke (CS) for 4 and 6 months. Scale bar, 200 μm.

B, C  Mean chord length (MCL) and iBALT quantification of lung sections from the mice described in (A), following 4 months (B) or 6 months (C) of CS exposure. (B) **$P = 0.0002$ and ****$P < 0.0001$ (emphysema quantification) and **$P = 0.0095$ (iBALT quantification). (C) **$P = 0.0022$ (emphysema quantification), *$P = 0.0207$ (FA vs. CS, WT mice, iBALT quantification), and *$P = 0.0373$ (CS WT vs. CS *Ch25h*$^{-/-}$ mice, iBALT quantification).

D    Left: Representative immunofluorescence images of the three regions quantified, stained to detect CD45R (B cells, red), CD3 (T cells, green), and DAPI (blue). A, airway; V, vessel. Scale bar, 50 μm. Right: Quantification of iBALT localized on the airway, vessels, and septal area from the mice described in (A). Airway: ***$P = 0.0007$ (FA vs. 6 m CS, WT mice) and ***$P = 0.0008$ (6 m CS WT vs. 6 m CS *Ch25h*$^{-/-}$ mice). Vessel: *$P = 0.0446$ (FA vs. 6 m CS, WT mice) and *$P = 0.0356$ (6 m CS WT vs. 6 m CS *Ch25h*$^{-/-}$ mice). Septum: *$P = 0.0194$ (FA vs. 6 m CS, WT mice).

E    Flow cytometric analysis of whole lung single-cell suspensions from mice in (A), to detect CD69-positive CD19 and CD3 cells. Left: Example dot plots of FA- and CS-exposed mice. Right: Frequency of CD69-positive cells. CD69$^+$ CD19$^+$ cells: **$P = 0.0062$ and ****$P < 0.0001$. CD69$^+$ CD3$^+$ cells: *$P = 0.0230$ and ****$P < 0.0001$.

Data information: Data are mean ± SD. *P*-values determined by one-way ANOVA and Tukey's multiple comparisons test. Data are representative of two independent experiments (A–D) or one experiment (E) with four mice per FA group or six mice per CS group.

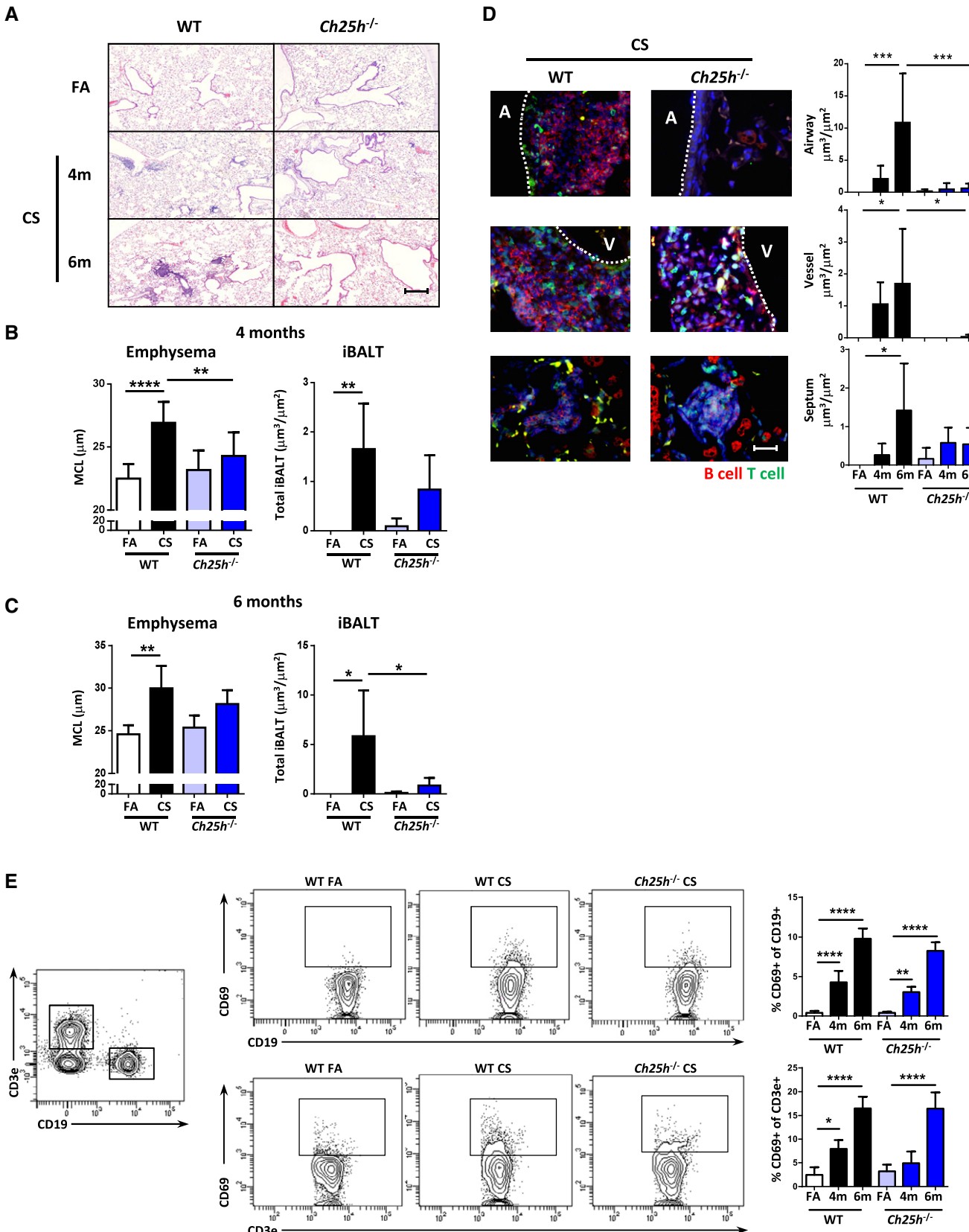

**Figure 2.**

## CH25H deficiency does not protect against iBALT-independent emphysema

Previous data demonstrate that professional APCs can express CH25H under inflammatory conditions (Park & Scott, 2010; Liu *et al*, 2013). These findings led us to interrogate whether differential cytokine secretion by resident alveolar or bone marrow-recruited macrophages (BMDM) deficient in CH25H were involved in emphysema development. Isolated alveolar macrophages significantly increased expression of *Ch25h* under both polarizing conditions (Fig EV5A), while BMDM from wild-type mice significantly increased expression of *Ch25h* only under M1-polarizing conditions

(Fig EV5B). Importantly, both alveolar and bone marrow macrophages from wild-type and *Ch25h$^{-/-}$* mice induced strong expression of *Tnfa* and *Il1b* under M1-polarizing conditions as well as *Irf4* and *Fizz1* as key transcription factors regulating M2 polarization (Satoh *et al*, 2010) after culturing with IL-4 (Fig EV5A and B). To extend these findings, we cultured bone marrow-derived DCs isolated from wild-type and *Ch25h$^{-/-}$* mice with LPS and found similarly increased expression of *Il12a*, *Tnfa,* and *Nos2* from both mice (Fig EV5C). In combination, this suggests that impaired cytokine secretion by professional APCs is not a contributing factor in CH25H-deficient mice. Furthermore, recent evidence revealed that depletion of alveolar macrophages ameliorated elastase-induced

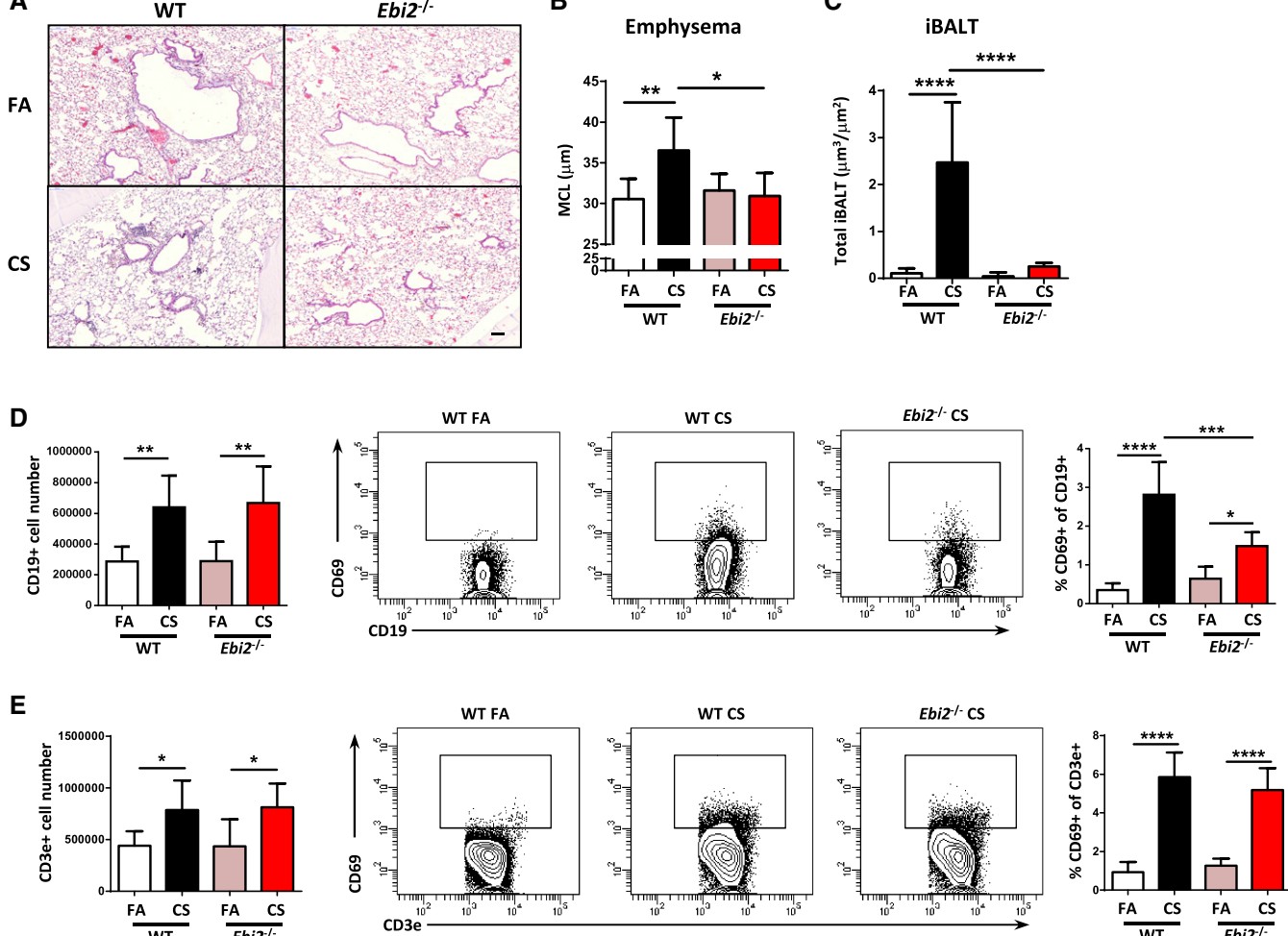

**Figure 3. EBI2-deficient mice are protected against iBALT formation and cigarette smoke-induced COPD.**

A  Representative H&E-stained lung from wild-type (WT) and EBI2-deficient (*Ebi2$^{-/-}$*) mice exposed to filtered air (FA) or cigarette smoke (CS) for 4 months. Scale bar, 100 μm.

B  Mean chord length (MCL) quantification of lung sections from the mice described in (A). *P = 0.0155 and **P = 0.0064.

C  Quantification of total lung iBALT from the mice in (A). ****P < 0.0001.

D, E  Flow cytometric analysis of whole lung single-cell suspensions from mice in (A), to detect CD19$^+$ and CD69$^+$ CD19$^+$ cells (D) and CD3$^+$ and CD69$^+$ CD3$^+$ cells (E). CD19$^+$ cells: **P = 0.0027 (FA vs. CS, WT mice) and **P = 0.0028 (FA vs. CS, *Ebi2$^{-/-}$* mice). CD69$^+$ CD19$^+$ cells: *P = 0.0232, ***P = 0.0003, and ****P < 0.0001. CD3$^+$ cells: *P = 0.0319 (FA vs. CS, WT mice) and *P = 0.0368 (FA vs. CS, *Ebi2$^{-/-}$* mice). CD69$^+$ CD3$^+$ cells: ****P < 0.0001.

Data information: Data are mean ± SD. *P*-values determined by one-way ANOVA and Tukey's multiple comparisons test. Data are from one experiment six mice per group (*Ebi2$^{-/-}$* CS), seven mice per group (WT CS or *Ebi2$^{-/-}$* FA), or eight mice per group (WT FA).

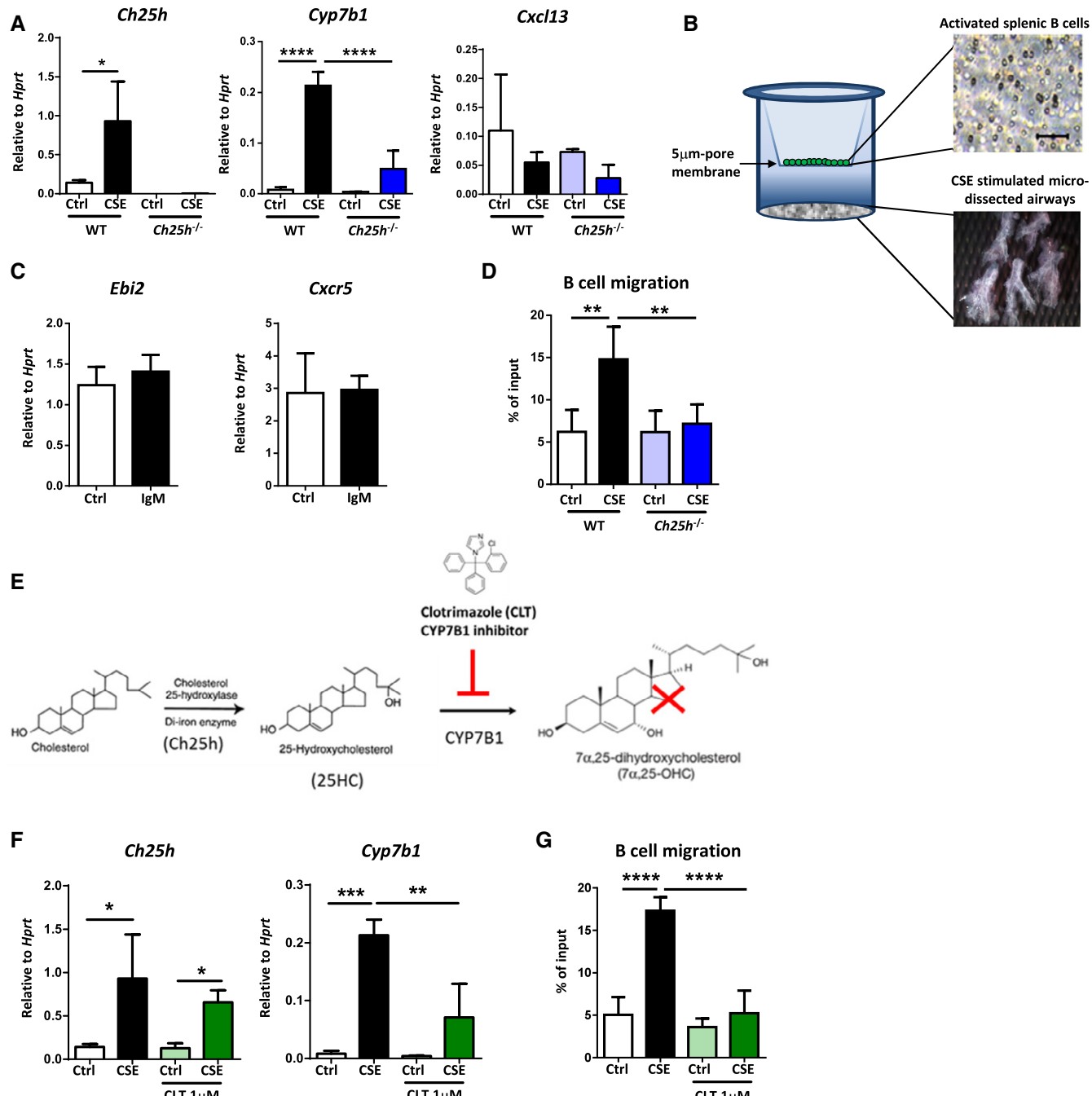

**Figure 4.    CH25H deficiency or attenuation of CYP7B1 activity with clotrimazole impairs B-cell migration toward CSE-treated airways *ex vivo*.**

A    *Ch25h*, *Cyp7b1*, and *Cxcl13* mRNA abundance from CSE-treated airways dissected from wild-type (WT) or CH25H-deficient (*Ch25h*$^{-/-}$) mice. *$P$ = 0.0400 and ****$P$ < 0.0001.

B    Schematic representation of *ex vivo* B-cell migration assay.

C    *Ebi2* and *Cxcr5* mRNA abundance from IgM cross-linked B cells isolated from the spleen of C57BL/6 mice.

D    Frequency of IgM cross-linked splenic B cells migrating toward medium from CSE-treated airways dissected from WT or *Ch25h*$^{-/-}$ mice. **$P$ = 0.0012 (Ctrl vs. CSE, WT airways) and **$P$ = 0.0035 (CSE WT vs. CSE *Ch25h*$^{-/-}$ airways).

E    Schematic representation of the metabolism of 7α,25-OHC from cholesterol.

F    *Ch25h* and *Cyp7b1* mRNA abundance from CSE-treated airways, in the absence or presence of 1 μM clotrimazole, dissected from C57BL/6 mice. *Ch25h*: *$P$ = 0.0297 (Ctrl vs. CSE, untreated airways) and *$P$ = 0.0112 (Ctrl vs. CSE, clotrimazole treated airways). *Cyp7b1*: **$P$ = 0.0056 and ***$P$ = 0.0008.

G    Frequency of IgM cross-linked splenic B cells migrating toward medium from CSE-treated airways, in the absence or presence of 1 μM clotrimazole, dissected from C57BL/6 mice. ****$P$ < 0.0001.

Data information: Data are mean ± SD. *P*-values determined by one-way ANOVA and Tukey's multiple comparisons test. All experiments repeated twice with three mice per group.

emphysema (Ueno *et al*, 2015), an iBALT-independent emphysema mouse model (Dau *et al*, 2015; Sarker *et al*, 2015). Therefore, we utilized this model to demonstrate that loss of CH25H in macrophages did not impact upon emphysema development. Both wild-type and $Ch25h^{-/-}$ mice developed a severe emphysema following elastase treatment, with no evidence of iBALT formation in either mice (Fig EV5D–H). These data further demonstrate that CH25H-deficient macrophages are not protective against elastase-induced emphysema, implying the role of CH25H in iBALT-mediated COPD pathogenesis.

### Oxysterols guide B-cell migration to the airways

7α,25-OHC guides B-cell positioning in secondary lymphoid tissue (Yi *et al*, 2012). To address the role of airway-specific CH25H in guiding B cells, we used microdissected airway trees (Yildirim *et al*, 2008) stimulated with cigarette smoke extract (CSE) *ex vivo*, which demonstrated increased expression of *Ch25h* and *Cyp7b1* in airways from wild-type mice, whereas $Ch25h^{-/-}$ mice did not express *Ch25h* and failed to increase the expression of *Cyp7b1* (Fig 4A). Isolated airways from both mice showed no differences in *Cxcl13* expression (Fig 4A). To further corroborate these findings, we utilized a novel *ex vivo* assay in which IgM cross-linked activated splenic B cells were tested for their ability to migrate toward the CSE-stimulated airway trees (Fig 4B). mRNA analysis of IgM cross-linked B cells revealed no change in the expression levels of the 7α,25-OHC receptor *Ebi2* (Hannedouche *et al*, 2011; Liu *et al*, 2011) and the CXCL13 receptor *Cxcr5* (Gunn *et al*, 1998; Legler *et al*, 1998; Fig 4C). Consistent with increased *Ch25h* and *Cyp7b1,* we observed a strong increase in the number of activated B cells migrating toward wild-type CSE-activated airways (Fig 4D).

To demonstrate that increased expression of CH25H- and CYP7B1-mediated oxysterol 7α,25-OHC guided B-cell migration, we cultured dissected wild-type airways in the presence of clotrimazole, a CYP7B1 inhibitor (Liu *et al*, 2011; Fig 4E). As expected, this treatment did not affect *Ch25h* levels, but was sufficient to reduce *Cyp7b1* mRNA expression (Fig 4F) and significantly impaired the ability of activated B cells to migrate toward CSE-treated airways (Fig 4G). These data suggested that 7α,25-OHC is guiding B-cell movement toward the airways. These results strongly indicate that

the CS-induced airway epithelial oxysterols are capable of driving B-cell migration, which contribute to iBALT generation on the airways in experimental COPD.

### Oxysterol inhibitor as a novel therapeutic target for COPD

As there are currently no clinical regimes to reverse the progression of emphysema (Vogelmeier *et al*, 2017), we evaluated whether inhibiting the generation of 7α,25-OHC could alleviate established experimental COPD when clotrimazole was administrated as a therapeutic dosing strategy (Fig 5A). Wild-type mice with pulmonary inflammation following two months chronic CS exposure (comparable with human COPD GOLD stages 0–1) were treated with clotrimazole during months two to four, leading to significantly reduced iBALT formation (Fig 5B and D) and lack of emphysema development (Fig 5C). At the cellular level, we observed reduced macrophage and lymphocyte cell numbers in BAL fluid of these clotrimazole-treated CS-exposed mice (Fig 5E). Importantly, wild-type mice exposed to CS for 4 months (comparable with human COPD GOLD stages 1–2), presenting clear signs of iBALT formation and COPD (Fig 2A–D and 5B–D), then treated with clotrimazole, showed attenuated iBALT and emphysema (Fig 5F–H). There was also reduced macrophage cell numbers in the BAL fluid of late clotrimazole-treated CS-exposed mice (Fig 5I). Taken together, these data suggest that inhibiting the generation of 7α,25-OHC with clotrimazole not only prevents iBALT formation, but is able to disrupt established iBALT and attenuate experimental CS-induced COPD.

## Discussion

This study reveals a role for oxysterol metabolism in guiding iBALT generation to the airways during COPD pathogenesis. Mice deficient in CH25H, an enzyme crucial for the metabolism of cholesterol toward the oxysterol 7α,25-OHC, or EBI2, the main receptor of 7α,25-OHC, did not generate iBALT in their lungs following exposure to chronic CS and were protected against the development of COPD. We also demonstrated that COPD patients and CS-exposed mice significantly upregulated CH25H and CYP7B1 expression in airway epithelial cells, and this was sufficient to promote B-cell

---

**Figure 5.  Clotrimazole protects against and reverses cigarette smoke-induced COPD.**

A    Schematic representation of clotrimazole therapeutic strategies.

B    Representative H&E-stained lung from C57BL/6 mice exposed to filtered air (FA) or cigarette smoke (CS) for 4 months and treated with clotrimazole (i.p. 80 mg/kg 3 times per week) from months 2 to 4 or oil controls (Early therapeutic group). Scale bar, 200 μm.

C    Mean chord length (MCL) quantification of lung sections from mice in (B). *$P$ = 0.0242 and ****$P$ < 0.0001.

D    Quantification of iBALT localized on the airway, vessels, and septal area from mice described in (B). *$P$ = 0.0249 (FA vs. CS, oil-treated mice) and *$P$ = 0.0116 (CS oil vs. CS clotrimazole-treated mice).

E    Bronchoalveolar lavage fluid (BALF) total and differential cell counts from mice in (B). Total cells: ***$P$ = 0.0003 and ****$P$ < 0.0001. Macrophages: **$P$ = 0.0033, ***$P$ = 0.0009 and ****$P$ < 0.0001. Neutrophils: ****$P$ < 0.0001. Lymphocytes: **$P$ = 0.0011 and ****$P$ < 0.0001.

F    Representative H&E-stained lung from C57BL/6 mice exposed to filtered air (FA) or cigarette smoke (CS) for 6 months and treated with clotrimazole (i.p. 80 mg/kg 3 times per week) from months 4 to 6 or oil controls (Late therapeutic group). Scale bar, 200 μm.

G    Mean chord length (MCL) quantification of lung sections from mice in (F). **$P$ = 0.0039.

H    Quantification of iBALT localized on the airway, vessels, and septal area from mice described in (F). Airway: *$P$ = 0.0101. Vessel: *$P$ = 0.0236. Septal: *$P$ = 0.0238.

I    BALF total and differential cell counts from mice in (F). Total cells: **$P$ = 0.0011 and ***$P$ = 0.0007. Macrophages: **$P$ = 0.0047. Neutrophils: **$P$ = 0.0012 and ***$P$ = 0.0004.

Data information: Data are mean ± SD. $P$-values determined by one-way ANOVA and Tukey's multiple comparisons test. Data are representative of two independent experiments with four mice per FA group, six mice per CS group (B–E), or five mice per CS group (F–I).

migration. Furthermore, activated B cells failed to migrate *ex vivo* toward CS-stimulated airways from *Ch25h*$^{-/-}$ or wild-type mice where oxysterol synthesis had been blocked. Finally, inhibition of

the oxysterol pathway, using the CYP7B1 inhibitor clotrimazole, resolved iBALT formation and attenuated CS-induced emphysema *in vivo* in a therapeutic approach.

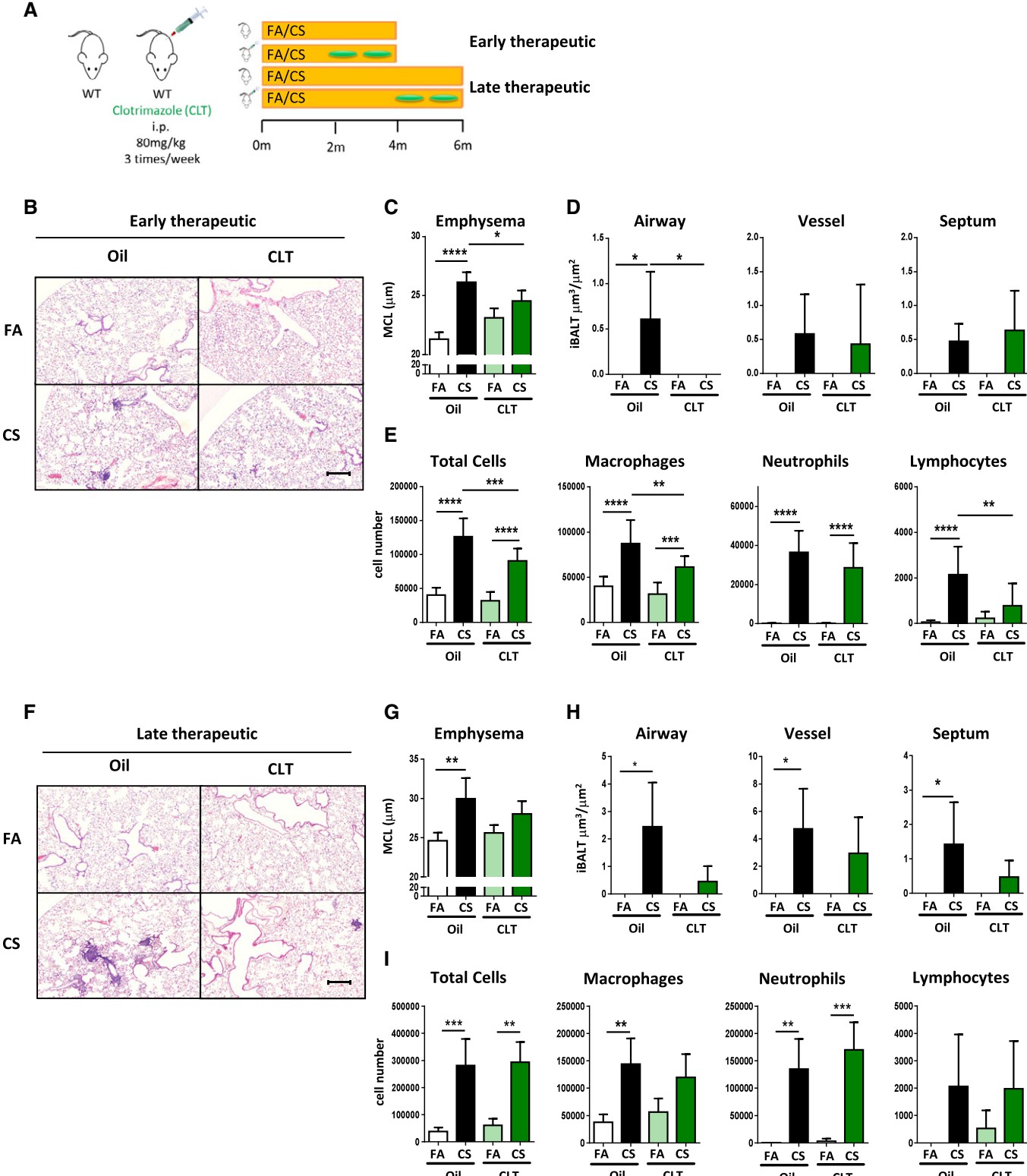

Figure 5.

Our data suggest that B cells are organized into iBALT structures upon the bronchus because of local secretion of 7α,25-OHC by the airway epithelial cells, akin to 7α,25-OHC generated by lymphoid stromal cells guiding activated B-cell movement during humoral responses (Yi et al, 2012). We do not exclude the possibility that T-cell migration is also disrupted, especially since EBI2 is expressed by CD4 T cells and activated T-cell positioning in the outer T zone of lymphoid follicles is directed by 7α,25-OHC (Li et al, 2016). However, we recently demonstrated that B cell-deficient mice were protected from CS-induced COPD despite normal function of CD4 T cells (John-Schuster et al, 2014). Notably, lymphocytes are recruited to the lung equivalently in wild-type mice and in mice where the oxysterol axis had been disturbed (Ch25h$^{-/-}$ and Ebi2$^{-/-}$ mice). Levels of Cxcl13, Cxcl9, Ccl19, and Ccl21 were increased in these mice following CS exposure equivalent to their wild-type counterparts, supporting our hypothesis.

The pro-inflammatory oxysterol 27-hydroxycholesterol (27-OHC) has also been demonstrated in the airways of COPD patients (Kikuchi et al, 2012) and recently described to be involved in splenic DC positioning and homeostasis (Lu et al, 2017). 27-OHC is synthesized by CYP7B1-driven metabolism of 27-hydroxycholesterol (Lu et al, 2017), and its synthesis will therefore be blocked by clotrimazole treatment. 27-OHC is, however, a poor chemokine for B cells (Liu et al, 2011), and as iBALT failed to form in Ch25h$^{-/-}$ mice following chronic CS exposure, this would suggest that 7α,25-OHC is the major EBI2 ligand for iBALT generation. It has also been reported that 27-OHC is able to accelerate senescence of both fibroblasts and airway epithelial cells (Hashimoto et al, 2016) and that 25-hydroxycholesterol may promote fibroblast-mediated tissue remodeling through NF-κB signaling (Ichikawa et al, 2013), suggesting that impairing oxysterol metabolism may have additional direct effects on lung tissue regeneration. However, we show here that Ch25h$^{-/-}$ mice are not protected against iBALT-independent emphysema development and that B cell-deficient mice upregulate oxysterol synthesizing enzymes after chronic CS exposure similar to their wild-type counterparts, yet are still protected against COPD development (John-Schuster et al, 2014).

Inducible bronchus-associated lymphoid tissue size and number correlates with the severity of COPD in patients (Hogg et al, 2004; Polverino et al, 2015), suggesting a therapeutic opportunity, especially considering current therapeutic regimes against COPD do not reverse the progression of emphysema (Tanaka et al, 2013; Vogelmeier et al, 2017). We clearly show that mice treated with the CYP7B1 inhibitor clotrimazole, resolved iBALT formation, and attenuated CS-induced emphysema in vivo. Based on our findings, we speculate that disruption of iBALT generation through the targeting of oxysterols, rather than complete B-cell depletion, particularly as a recent rituximab trial in COPD patients failed because of increased risk of infectious complications (Brusselle et al, 2009), opens new therapeutic strategies on a broader perspective for other diseases associated with tertiary lymphoid organs beyond COPD, such as pulmonary hypertension, cancer, transplant rejection, and autoimmunity (Pitzalis et al, 2014).

# Materials and Methods

## Transcriptomic data analysis

Microarray data were obtained from data series held at the NCBI Gene Expression Omnibus (GEO) database (Barrett et al, 2013).

Mouse whole lung data of filtered air versus chronic CS-exposed animals were obtained from accession GSE52509 (John-Schuster et al, 2014), and human data comparing gene expression in the small airway epithelial cells of COPD patients with healthy non-smoking controls from accession GSE11784 (Tilley et al, 2011). Data were analyzed using the GEO web tool GEO2R (Barrett et al, 2013) with default settings, whose back end uses Bioconductor (Gentleman et al, 2004) R packages to transform and analyze the data, to generate the log$_2$ transformed expression values for each gene relative to filtered air or healthy controls. Gene Ontology pathway analysis was undertaken on GEO2R gene expression data with a $P < 0.05$ using the web-based gene set analysis toolkit WebGestalt (Wang et al, 2013). A heat map of selected genes (log$_2$ transformed expression values, with a $P < 0.05$, as calculated by GEO2R) taken from the Gene Ontology pathway analysis was generated by Genesis software (Sturn et al, 2002; Release 1.7.7, Institute for Genomics and Bioinformatics, Graz University of Technology).

Gene set enrichment analysis (GSEA) software from the Broad Institute (http://www.gsea-msigdb.org/gsea/index.jsp; Mootha et al, 2003; Subramanian et al, 2005) was used to determine the enrichment of Reactome gene lists obtained from the GSEA-Molecular Signatures Database for total TLR- and TLR4-specific signaling. Data from the series matrix file comparing gene expression in the small airway epithelial cells of COPD patients with healthy smoking controls (accession GSE11784) were downloaded from the NCBI GEO database.

## Human lung tissue

Lung core samples of emphysematous and healthy regions from the same explanted lungs of COPD patients undergoing lung transplantation were provided by Dr. Stijn Verleden (University of Leuven, Belgium) following ethical approval of the University of Leuven Institutional Review Board (ML6385). All participants provided written consent and experiments conformed to the principles set out in the Declaration of Helsinki. Patient demographics are highlighted in Table 1. Immediately following transplant, lungs were air-inflated at

Table 1. Demographics and clinical characteristics of COPD transplant patients (mean ± SEM). Lung core samples were histologically separated into non-emphysematous and emphysematous tissue.

|  | COPD |
| --- | --- |
| Subjects (n) | 16 |
| Mean age years | 57.06 ± 1.23 |
| Sex |  |
| Male | 7 |
| Female | 9 |
| Height (m) | 1.65 ± 0.02 |
| Weight (kg) | 59.44 ± 3.18 |
| Smoking (packs/year) | 39.00 ± 7.55 |
| FEV1 (%) | 34.38 ± 5.55 |
| FVC (%) | 82.73 ± 5.84 |
| FEV1/FVC (%) | 31.01 ± 5.07 |

FEV1, Forced expiratory volume in the first second; FVC, Forced vital capacity.

10 cm $H_2O$ pressure and fixed using constant pressure in the fumes of liquid nitrogen. Afterward lungs were sliced using a band saw and sampled using a core bore. Upon receipt, lung cores were portioned for fixation in 4% paraformaldehyde followed by paraffin embedding and total RNA isolation (peqGOLD Total RNA Kit, Peqlab).

### Primary bronchial epithelial cell culture

Primary bronchial epithelial cell cultures were obtained from smokers with and without COPD, who had lobectomy or pneumonectomy for lung cancer or other reasons at Gaziantep University Hospital (Turkey), using an explant cell culture technique (Devalia *et al*, 1990; Bayram *et al*, 1998). The demographic characteristics of patients are given in Table 2. The study was approved by The Ethics Committee of Gaziantep University, informed written consents were taken from study volunteers, and experiments conformed to the principles set out in the Declaration of Helsinki. Total cellular RNA was isolated using commercially available kits (QIAcube, Qiagen).

### RNAseq analysis

Total RNA of human lung derived from COPD explants or healthy donor controls was sequenced using the Illumina system (HiSeq 2000) by the company GATC Biotech.AG (Konstanz, Germany). Raw data analysis was performed by Genomatix Software GmbH. Expression values were calculated as RPKM (Reads per kilobase million per mapped reads) for all loci available from reads uniquely aligned to the human genome. In this study, we only present the expression of *CH25H*.

### Mice

B6.129S6-*Ch25h*[tm1Rus]/J (*Ch25h*[−/−]) mice were obtained from The Jackson Laboratory and B6N(Cg)-*Gpr183*[tm1.1(KOMP)Vlcg]/J (*Ebi2*[−/−]) from the KOMP Repository, University of California Davis. Age-matched female C57BL/6J mice and B cell-deficient B6.129S2-*Igh-6*[tm1Cgn] (μMT) mice were obtained from Charles River Laboratories. Mice were housed under specific pathogen-free conditions at a constant temperature and humidity with a 12-hour light cycle and allowed food and water *ad libitum*. All animal experiments were performed according to strict governmental and international guidelines and were approved by the local government for the administrative region of Upper Bavaria, Germany.

### Mouse COPD models and treatment

Eight- to 12-week-old female mice were used in all experiments. For the CS-induced COPD model, mice were exposed to 100% mainstream CS (John *et al*, 2014) at a particle concentration of 500 mg/m$^3$, generated from 3R4F research cigarettes (Filter removed, Tobacco Research Institute, University of Kentucky), for 50 min twice/day, 5 days/week for 4 or 6 months. Mice exposed to filtered air were used as controls. For the elastase-induced iBALT-independent COPD model, mice were instilled oropharyngeally with a single application of 80 U/kg body weight porcine pancreatic elastase (PPE; Sigma-Aldrich) in 80 μl PBS. Mice treated with 80 μl PBS were included as controls. For CYP7B1 inhibition, clotrimazole (Sigma-Aldrich) 10 mM in DMSO was further diluted in corn oil (Mazola, Unilever) and applied i.p. at a dose of 80 mg/kg body weight 3 times/week for 2 months. Mice were treated after 2 or 4 months of CS exposure and smoked for a further 2 months in parallel with the clotrimazole treatment.

### Lung function analysis

Mice were anaesthetized with ketamine–xylazine, tracheostomized, cannulated, and the diffusing capacity for carbon monoxide (DFCO) calculated (Fallica *et al*, 2011). Briefly, 0.8 ml mixed gas (0.5% Ne, 21% $O_2$, 0.5% CO and 78% $N_2$) was instilled into the mice lungs and withdrawn 2 s later for analysis on a 3000 Micro GC Gas Analyzer (Infinicon). DFCO was calculated as $1-(CO_1/CO_0)/(Ne_1/Ne_0)$ where 0 and 1 refer to the gas concentration before and after instillation, respectively. Respiratory function was analyzed using a forced pulmonary maneuver system (Vanoirbeek *et al*, 2010; Buxco Research Company, Data Sciences International) running FinePointe Software (version 6, Data Sciences International) and the quasistatic PV maneuver protocol.

### Bronchoalveolar lavage

After lung function analysis, lungs were lavaged with 3 × 500 μl aliquots of sterile PBS (Gibco, Life Technologies) supplemented with Complete Protease Inhibitor Cocktail tablets (Roche Diagnostics). Cells were pelleted at 400 *g* for 20 min and resuspended in 500 μl RPMI-1640 medium (Gibco, Life Technologies) for the total cell count using a Neubauer Chamber. Cytospins of the cell suspension were then prepared and stained using May-Grünwald-Giemsa for differential cell counting (200 cells/sample) using morphological criteria. Bronchoalveolar lavage fluid was retained for mass spectrometry analysis.

### Mouse lung processing

The two right lower lung lobes were snap-frozen in liquid nitrogen, homogenized and total RNA isolated (peqGOLD Total RNA Kit, Peqlab). The right upper two lobes were dissociated into single-cell suspensions in PBS supplemented with 0.1% FCS and 2 mM EDTA

**Table 2. Demographics and clinical characteristics of study subjects for the primary bronchial epithelial cell cultures (mean ± SEM).**

|  | Smokers | COPD |
|---|---|---|
| Subjects (*n*) | 10 | 11 |
| Mean age years | 61.10 ± 12.75 | 65.82 ± 8.49 |
| Smoking (packs/year) | 32.00 ± 11.83 | 49.55 ± 26.12** |
| FEV1 (%) | 92.80 ± 8.50 | 75.64 ± 15.58* |
| FVC (%) | 98.30 ± 12.07 | 92.91 ± 14.41 |
| FEV1/FVC (%) | 76.00 ± 3.68 | 63.36 ± 6.32** |
| GOLD (mean, min-max) | NA | 1.73 (1,3) |

FEV1: Forced expiratory volume in the first second; FVC: Forced vital capacity.
*$P < 0.01$, **$P < 0.001$ COPD versus smokers (Mann–Whitney test).

using the lung dissociation kit and gentleMACS Dissociator from Miltenyi Biotec for flow cytometry analysis. The left lung was inflation fixed with 6% paraformaldehyde under a constant pressure of 20 cm and then embedded into paraffin.

### Quantitative real-time RT–PCR

1 μg RNA was reverse transcribed using Random Hexamers and MuLV Reverse Transcriptase (Applied Biosystems) or by the Precision Reverse Transcription Kit (Qiagen). Gene expression was analyzed using SensiFAST SYBR Hi-ROX Kit (Bioline) on a StepOnePlus 96-well Real-Time PCR System (Applied Biosystems) or a

RG-600 model RT–PCR machine (Corbett Research). Primer sequences can be found in Table 3. Expression of each gene was calculated relative to the housekeeping gene *HPRT1* or *Hprt1* as $2^{-\Delta C_t}$.

### Immunofluorescence staining

3-μm sections from mouse left lung or human core samples were stained as described (John-Schuster *et al*, 2016). Briefly, sections were deparaffinized, rehydrated, and heat-induced epitope retrieval undertaken using HIER Citrate Buffer (pH 6.0, Zytomed Systems). Sections were blocked using 5% BSA in PBS and then incubated

**Table 3.    Primer sequences used for the quantitative real-time RT–PCR.**

| Gene | Forward primer | Reverse Primer |
|---|---|---|
| CH25H | CTC TAC CAG CAT GTG ATG TTT GT | CAT GTC GAA GAG TAG CAG GCA |
| CXCL8 | GGC TCT CTT GGC AGC CTT C | GGT TTG GAG TAT GTC TTT ATG CAC |
| CXCL13 | CAA GTC AAT TGT GTG TGT GGA | GGG AAT CTT TCT CTT AAA CAC TGG |
| HPRT1 | AGG AAA GCA AAG TCT GCA TTG TT | GGT GGA GAT GAT CTC TCA ACT TTA A |
| TLR4 | AGA CCT GTC CCT GAA CCC TAT | CGA TGG ACT CTA AAC CAG CCA |
| Adgre1 | CTC TGT GGT CCC ACC TTC AT | GAT GGC CAA GGA TCT GAA AA |
| Arg1 | GGA ACC CAG AGA GAG CAT GA | TTT TTC CAG CAG ACC AGC TT |
| Ccl19 | TGG GAA CAT CGT GAA AGC CT | GTG GTG AAC ACA ACA GCA GG |
| Ccl21 | CGG CTG TCC ATC TCA CCT AC | AGG GAA TTT TCT TCT GGC TGT |
| Ch25h | GAC CTT CTT CGA CGT GCT GA | CCA CCG ACA GCC AGA TGT TA |
| Cxcl1 | CCG AAG TCA TAG CCA CAC | GTG CCA TCA GAG CAG TCT |
| Cxcl13 | TCT CTC CAG GCC ACG GTA TTC T | ACC ATT TGG CAC GAG GAT TCA C |
| Cxcl9 | CGA GGC ACG ATC CAC TAC AA | AGG CAG GTT TGA TCT CCG TT |
| Cxcr5 | TGG ATG ACC TGT ACA GGG AAC TG | CGG TGC CTC TCC AGG ATT AC |
| Cyp27a1 | GGA GGG CAA GTA CCC AAT AA | TTC AGC AGC CTC TGT TTC AA |
| Cyp7b1 | GGA GCC ACG ACC CTA GAT G | GCC ATG CCA AGA TAA GGA AGC |
| Ebi2 | ATG GCT AAC AAT TTC ACT ACC CC | ACC AGC CCA ATG ATG AAG ACC |
| Fizz1 | TGC CAA TCC AGC TAA CTA TCC C | ACG AGT AAG CAC AGG CAG TT |
| Gmcsf | ATG CCT GTC ACG TTG AAT GA | CCG TAG ACC CTG CTC GAA TA |
| Hprt1 | AGC TAC TGT AAT GAT CAG TCA ACG | AGA GGT CCT TTT CAC CAG CA |
| Hsd3b7 | AGT GGT GGG GCC TAA CAT CA | CTG CTC AGC AAG GGC TTT AC |
| Il12p35 | ACT AGA GAG ACT TCT TCC ACA ACA AGA G | GCA CAG GGT CAT CAT CAA AGA C |
| Il1a | AGC GCT CAA GGA GAA GAC | CTG TCA TAG AGG GCA GTC C |
| Il1b | CAA CCA ACA AGT GAT ATT CTC CAT G | GAT CCA CAC TCT CCA GCT GCA |
| Il6 | GTT CTC TGG GAA ATC GTG GA | TGT ACT CCA GGT AGC TAT GG |
| Irf4 | AAA GGC AAG TTC CGA GAA GGG | CTC GAC CAA TTC CTC AAA GTC A |
| Lta | TCC ACT CCC TCA GAA GCA CT | AGA GAA GCC ATG TCG GAG AA |
| Ltb | TAC ACC AGA TCC AGG GGT TC | ACT CAT CCA GCG CCT ATG A |
| Ltbr | AAG CCG AGG TCA CAG ATG AAA | CGA GGG GAG GAA GTG TTC TG |
| Mcp1 | CTT CTG GGC CTG CTG TTC A | CCA GCC TAC TCA TTG GGA TCA |
| Mmp12 | TGT ACC CCA CCT ACA GAT ACC TTA | CCA TAG AGG GAC TGA ATG TTA CGT |
| Nos2 | CGG CAA ACA TGA CTT CAG GC | GCA CAT CAA AGC GGC CAT AG |
| Timp1 | CAC TGA TAG CTT CCA GTA AGG CC | CTT ATG ACC AGG TCC GAG TTG C |
| Tnfa | CAC CAC GCT CTT CTG TCT | GGC TAC AGG CTT GTC ACT C |

overnight at 4°C with primary antibody, followed by 1 h with secondary antibody and counterstained with DAPI (1:4,000, Sigma-Aldrich), mounted in fluorescent mounting medium (Dako), and imaged with a fluorescent Olympus BX51 microscope running cellSens software (Version 1.14, Build 14116, Olympus). Primary antibodies: rat IgG2a anti-mouse CD45r (1:50, clone: RA3-6B2, BD Biosciences), rabbit IgG1 anti-mouse CD3 (1:300, Cat. No. C7930, Sigma-Aldrich), mouse IgG2b anti-human/mouse CH25H (1:500, Cat. No. ab76478, Abcam). Secondary antibodies: Alexa Fluor 488 conjugated goat anti-mouse IgG antibody (1:300, Cat. No. A11001, ThermoFisher Scientific), Alexa Fluor 488 conjugated goat anti-rabbit IgG antibody (1:300, Cat. No. A11008, ThermoFisher Scientific), Alexa Fluor 568 conjugated goat anti-rat IgG antibody (1:300, Cat. No. A11077, ThermoFisher Scientific).

## Immunohistochemical staining

3-μm sections from human core samples or mouse lung were deparaffinized, rehydrated, and then treated with 1.8% (v/v) $H_2O_2$ solution (Sigma-Aldrich) to block endogenous peroxidase. Heat-induced epitope retrieval was performed in HIER citrate buffer (pH 6.0, Zytomed Systems) in a decloaking chamber (Biocare Medical). To inhibit non-specific binding of antibodies, sections were treated with a blocking antibody (Biocare Medical). Human sections were incubated at 4°C overnight with a rabbit anti-TLR4 primary antibody (1:50, Cat. No. ab13556, Abcam), followed by 1 h with an anti-rabbit HRP-conjugated secondary antibody (Biocare Medical). Signals were amplified by adding chromogen substrate 3,3′-diaminobenzidine (DAB; Biocare Medical). Mouse sections were incubated at 4°C overnight with a rabbit anti-galectin-3 primary antibody (1:100, Cat. No. sc-20157, Santa Cruz Biotechnology), followed by 1 h with a Rabbit-on-Rodent AP-Polymer (Biocare Medical). Signals were amplified by adding chromogen substrate Vulcan fast red (Biocare Medical). All sections were counterstained with hematoxylin (Sigma-Aldrich), dehydrated, and mounted.

## Quantitative morphometry

H&E-stained tissue sections were analyzed by design-based stereology using an Olympus BX51 light microscope equipped with the new Computer Assisted Stereological Toolbox (newCAST, Visiopharm) as described (John-Schuster et al, 2014), by readers blinded to the study groups. Briefly, for mean chord length (MCL) measurements, 20 frames were selected randomly across multiple sections by the software, using the ×20 objective, and superimposed by a line grid and points. The intercepts of lines on alveolar wall ($I_{septa}$) and points localized on air space ($P_{air}$) were counted and calculated as $MCL = \sum P_{air} \times L(p) / \sum I_{septa} \times 0.5$, where $L(p)$ is the line length per point. The volume of inflammation (V) was quantified in 50 frames, using the ×40 objective, by counting points hitting inflammatory cell zones ($P_{inflam}$). For calculation, the $P_{inflam}$ were referenced to intercepts of lines with both airways and vessels ($I_{airway/vessel}$): $V = \sum P_{inflam} \times L(p) / \sum I_{airway/vessel}$. Further, airway-, vessel-, or septum-associated inflammation quantification was classified by the location of the inflammation and was calculated referring to intercept of lines with airway, vessel, or both, respectively.

## Flow cytometry

$10^6$ cells from filtered single-cell lung suspensions were blocked with purified anti-mouse CD16/CD32 (Clone: 93, eBioscience) before incubating for 30 min on ice with antibody cocktails. After washing and re-suspending in MACS buffer, cells were analyzed on a BD FACSCanto II flow cytometer (BD Biosciences) and BD FACS-Diva software. B-cell and T-cell staining was performed with: APC-conjugated anti-CD19 (clone: 6D5, Miltenyi Biotec), APC-Vio770-conjugated anti-CD3e (clone: 17A2, Miltenyi Biotec), PE-Vio770-conjugated anti-CD22 (clone: Cy34.1, Miltenyi Biotec), PE-conjugated anti-CD80 (clone: 16-10A1, Miltenyi Biotec), PerCP-Vio700-conjugated anti-MHCII (clone: M5/114.15.2, Miltenyi Biotec), VioGreen-conjugated anti-CD69 (clone: H1.2F3, Miltenyi Biotec), FITC-conjugated anti-IgG (Biolegend), VioBlue-conjugated anti-GL7 (Biolegend). For the macrophage profile: VioGreen-conjugated anti-CD45 (clone: 30F11, Miltenyi Biotec), APC-Vio770-conjugated anti-Ly6C (clone: 1G7.G10, Miltenyi Biotec), VioBlue-conjugated anti-Ly6G (clone: 1A8, Miltenyi Biotec), FITC-conjugated anti-MHCII (clone: M5/114.15.2, Miltenyi Biotec), PerCP-Vio700-conjugated anti-F4/80 (clone: REA126, Miltenyi Biotec), PE-conjugated anti-CD11b (clone: M1/70.15.11.5, Miltenyi Biotec), APC-conjugated anti-CD11c (clone: N418, Miltenyi Biotec), PE-Vio770-conjugated anti-CD64 (clone: REA286, Miltenyi Biotec).

## Microdissection of airways

Middle and distal airways from C57BL/6J and $Ch25h^{-/-}$ mice were isolated and incubated ex vivo as described (Yildirim et al, 2008). Briefly, after sacrifice by a ketamine–xylazine over dose, the trachea was cannulated, the lungs removed from the thorax and infused with 1% low-melting agarose dissolved in 1:1 Ham's F12 nutrient medium (Sigma-Aldrich) and distilled water (Gibco, Life Technologies). Airways were dissected under a microscope (Zeiss) from the left lung after the agarose had solidified on ice for 30 min. The isolated airways were washed and cultured in airway epithelial cell medium (PromoCell) at 37°C, 5% $CO_2$.

## B-cell isolation and migration assay

B cells were purified from the spleens of C57BL/6J mice by negative selection (B cell Isolation Kit, mouse, Miltenyi Biotec). For the migration assay, primary mouse airways were isolated 1 day prior and treated with 10% CSE in airway epithelial cell culture medium (PromoCell) or culture medium alone for 24 h. To inhibit CYP7B1, clotrimazole in DMSO was diluted with culture medium or combined with 10% CSE to a final concentration of 1 μM. The supernatants were transferred as conditioned medium to the lower well of 24-well transwell plates (Permeable Polycarbonate Membrane Inserts, Corning, Fisher Scientific), for inducing B-cell migration, while the airway samples were snap-frozen in liquid nitrogen for RNA isolation. Freshly isolated B cells at $2.5 \times 10^6$/ml in 100 μl were activated by unconjugated AffiniPure F(ab')$_2$ Fragment Goat anti-mouse IgM, μchain-specific antibody (1.3 μg/ml, 115-006-020, Jackson Immunoresearch Laboratories) for 1 h at 37°C in 5.0 μm pore-sized transwell inserts (Permeable Polycarbonate Membrane Inserts, Corning, Fisher Scientific). Transwell inserts were then placed into the wells of conditioned medium and

incubated for 3 h at 37°C. Migrated B cells were collected and counted by Neubauer Chamber and reported as percentage of input.

## Cigarette smoke extract preparation

Cigarette smoke extract was generated by bubbling smoke from three research cigarettes (3R4F, Tobacco Research Institute, University of Kentucky) through 30 ml of airway epithelial cell culture medium (PromoCell) at a puffing speed equating to one cigarette every 5 min, in a closed environment with limited air flow. This solution was taken as 100% CSE.

## Isolation and stimulation of professional APCs

Primary alveolar macrophages were isolated from the lungs of C57BL/6J and *Ch25h*$^{-/-}$ mice by BAL with 10 washes of 1 ml PBS (Gibco, Life Technologies). Cells were pelleted at 400 *g* for 20 min and resuspended in complete RPMI-1640 medium supplemented with 10% fetal bovine serum, 50 μM β-mercaptoethanol, and 100 U/ml penicillin and streptomycin (all from Gibco, Life Technologies). $5 \times 10^4$ cells in 1 ml were seeded in 24-well plates and allowed to adhere for 1 h. Non-adherent cells were removed by washing twice with PBS. To generate bone marrow-derived macrophages (BMDM) and DCs (BMDC), bone marrow was flushed from femurs and tibias of C57BL/6J and *Ch25h*$^{-/-}$ mice with RPMI-1640 medium. Cells were disaggregated by passing through a 40-μm mesh and cultured in complete RPMI-1640 medium supplemented with 5% fetal bovine serum, 50 μM β-mercaptoethanol, and 100 U/ml penicillin and streptomycin at a concentration of $1 \times 10^6$ cells/ml in 24-well plates. For BMDMs, the medium was supplemented with 20 ng/ml murine recombinant M-CSF (ImmunoTools), and for BMDCs, the medium was supplemented with 20 ng/ml murine recombinant GM-CSF (ImmunoTools) and cultured at 37°C, 5% $CO_2$. Cells were maintained by replacing the medium with fresh medium on alternate days ensuring removal of non-adherent cells. On day 6, adherent BMDMs were collected. For BMDCs, on day 7–8 adherent cells were harvested and resuspended at $1 \times 10^6$ cells/ml in 10 ml complete RPMI-1640 medium in 100-mm petri dishes and cultured for a further 24–48 h. The non-adherent, non-proliferating, maturing DCs were collected as they were released. Primary alveolar macrophages and BMDMs were polarized toward M1 by culturing with complete RPMI-1640 medium containing 1 μg/ml LPS (from *Escherichia coli* 0111:B4, Sigma-Aldrich) and 20 ng/ml recombinant murine IFNγ (ImmunoTools) for 24 h or an M2 phenotype with 20 ng/ml recombinant murine IL-4 (ImmunoTools) for 24 h. BMDCs were stimulated with 1 μg/ml LPS for 24 h.

## Bronchial epithelial cell lines

The human bronchial epithelial cell lines BEAS-2B (ATCC CRL-9609) and 16-HBE (Cozens *et al*, 1994) were maintained in airway epithelial cell medium (PromoCell) supplemented with 10% fetal bovine serum and 100 U/ml penicillin and streptomycin (all from Gibco, Life Technologies) at 37°C, 5% $CO_2$. Cells were seeded and grown to confluence over 48 h in 24-well plates before stimulation with LPS (from *E. coli* 0111:B4, Sigma-Aldrich), CSE or recombinant human TNF-α (PeproTech), at the concentrations described in the figure legends.

### The paper explained

**Problem**

Long-term environmental exposure to toxic gases and particles, in particular cigarette smoke, can result in chronic obstructive pulmonary disease (COPD), currently the third leading cause of death worldwide. It manifests as chronic bronchitis, small airway remodeling, and emphysema, resulting in progressive and largely irreversible airflow limitation and impaired gaseous exchange. Currently, the only available therapies aim at symptom management and do not reverse disease progression. There is strong evidence that advanced stages of COPD are driven by the generation of inducible bronchus-associated lymphoid tissue (iBALT). Yet, we do not know how iBALT gets organized upon the bronchi and if its disruption can reverse COPD progression.

**Results**

We have demonstrated that oxysterols, metabolites of cholesterol, are critically involved in iBALT generation and the immune pathogenesis of COPD. In both, COPD patients and mouse models of the disease, we identified upregulated oxysterol enzymes in airway epithelial cells. Furthermore, mice genetically or pharmacologically deficient in the oxysterol pathway were protected from cigarette smoke-induced emphysema and iBALT formation.

**Impact**

This study provides valuable new insights into the mechanism of iBALT-driven COPD pathogenesis and highlights the oxysterol pathway as a potential therapeutic approach for COPD disease progression, and conceivably, the many other chronic diseases associated with tertiary lymphoid organ development.

## Analysis of 25-hydroxycholesterol

Determination of 25-hydroxycholesterol in cell culture supernatant and BALF was performed based on mass spectrometric methods previously described for different instrumentation (Honda *et al*, 2009; Huang *et al*, 2014). 25-hydroxycholesterol was derivatized, and the product was analyzed using ultra-high pressure liquid chromatography (UHPLC) coupled with high-resolution time-of-flight mass spectrometry (LC-HRTOF-MS). UHPLC separation was performed on a 1290 Infinity Binary LC-System using an Eclipse C-18, 1.8 μm, 50 × 2.1 mm I.D. analytical column (both from Agilent Technologies). Mass spectrometric detection was accomplished on a Citius™ High Resolution multi-reflection time-of-flight mass spectrometer (LC-HRT, Leco).

## Metabolomics analysis

The targeted metabolomics approach was based on LC-ESI-MS/MS and FIA-ESI-MS/MS measurements by Absolute*IDQ*™ p180 Kit (BIOCRATES Life Sciences AG) which has been described in detail (Zukunft *et al*, 2013). Frozen lung tissue was homogenized and extracted as described previously (Conlon *et al*, 2016; Zukunft *et al*, 2018). Mass spectrometric analyses were done on an API 4000 triple quadrupole system (Sciex Deutschland GmbH) equipped with a 1200 Series HPLC (Agilent Technologies) and a HTC PAL auto sampler (CTC Analytics) controlled by the software Analyst 1.5.1. Data evaluation for quantification of metabolite concentrations and quality assessment was performed with the MetIDQ™ software

package, which is an integral part of the AbsoluteIDQ™ Kit. Individual metabolite concentrations for each sample can be found in Dataset EV1, with a more detailed description of the methods found in Appendix Supplementary Methods.

## Statistical analysis

No statistical methods were used to predetermine sample size. GraphPad Prism (Version 6, GraphPad Software) was used for all statistical analysis. Data are presented as mean ± SD with sample size and number of repeats indicated in the figure legends. For comparison between two groups, statistical significance was analyzed with Student's *t*-test. For multiple comparisons, one-way ANOVA and Tukey's multiple comparisons test were used. $P < 0.05$ were considered significant.

**Expanded View** for this article is available online.

## Acknowledgements

The authors acknowledge the help of Christine Hollauer, Heike Bollig, and Maximilian Pankla. We thank Julia Scarpa, Werner Römisch-Margl, and Katharina Faschinger for metabolomics measurements performed at the Helmholtz Zentrum München, Genome Analysis Center, Metabolomics Core Facility. RZ, JL, XW, and AÖY thank the HGF for supporting the virtual Institute HICE.

## Author contributions

TMC, JJ, OE, and AÖY conceived the study and experimental design. TMC, JJ, RSJS, NFS, BS, GG, DT, XW, JG, KH, and MI performed experiments. SEV prepared patient lung core samples. DT and HB generated primary human bronchial epithelial cell cultures and undertook analysis. XW, RZ, and JL designed and analyzed oxysterol mass spectrometry measurements. CP and JA designed and analyzed targeted metabolomics. MI, JB, and MHA contributed to microarray analysis. SP and MS contributed to lung mycobiome analysis. TMC, JJ, RSJS, NFS, and AÖY analyzed and interpreted the data. TMC, JJ, OE, and AÖY wrote the manuscript. All authors read and edited the manuscript.

## Conflict of interest

The authors declare that they have no conflict of interest.

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
