## [Review Process File · EMBO Molecular Medicine]

Cholesterol metabolism promotes B cell positioning during immune pathogenesis of chronic obstructive pulmonary disease

Jie Jia, Thomas M. Conlon, Rim S. J. Sarker, Demet Taşdemir, Natalia F. Smirnova, Barkha Srivastava, Stijn E. Verleden, Gizem Güneş, Xiao Wu, Cornelia Prehn, Jiaqi Gao, Katharina Heinzlmann, Jutta Lintelmann, Martin Irmeler, Stefan Pfeiffer, Michael Schloter, Ralf Zimmermann, Martin Hrabé de Angelis, Johannes Beckers, Jerzy Adamski, Hasan Bayram, Oliver Eickelberg, Ali Önder Yildirim

Review timeline:

Submission date:	31 July 2017
Editorial Decision:	22 September 2017
Revision received:	22 December 2017
Editorial Decision:	9 February 2018
Revision received:	22 February 2018
Editorial Decision:	26 February 2018
Revision received:	8 March 2018
Accepted:	14 March 2018

Editor: Roberto Buccione/Céline Carret

Transaction Report:

1st Editorial Decision

22 September 2017

Thank you for the submission of your manuscript to EMBO Molecular Medicine. I apologise for the delay in reaching a decision. In fact, we experienced significant difficulties in securing expert and willing reviewers in part due to the overlapping holiday season.

Although I was hoping to obtain a third evaluation, I am now proceeding based on the two evaluations obtained so far as further delays cannot be justified.

You will see that although both Reviewers are appreciative your work and underline its potential interest, a few critical and partially overlapping concerns are raised.

Specifically, while reviewer 2 is globally positive, reviewer 1 appears rather reserved. The latter's concerns are chiefly on the lack of mechanistic insight into a number of events, namely 1) how iBALTs cause emphysema, 2) how does the regional upregulation of CH25H occur, 3) how does protection from emphysema occur in the Ch25h^{-/-} mice occur and 4) how does the antifungal drug clotrimazole impact on the lung microbiome and 5) how does clotrimazole reverse emphysema.

In conclusion, while publication of the paper cannot be considered at this stage, we are willing to consider a substantially revised manuscript provided however, that the Reviewers' concerns are fully addressed with additional experimentation where required. I will not be asking you to directly address with further experimentation items 1 and 5 specifically, although if you have the data available, I would encourage you to incorporate them into the manuscript. I do suggest however that you better investigate the regional upregulation of CH25H, how protection from emphysema occurs in the Ch25h^{-/-} mice and the mechanism of action of clotrimazole.

Please note that it is EMBO Molecular Medicine policy to allow a single round of revision only and that, therefore, acceptance or rejection of the manuscript will depend on the completeness of your responses included in the next, final version of the manuscript.

I understand that to address the above might entail a significant amount of time, additional work and experimentation and might be technically challenging, I would therefore understand if you chose to rather seek publication elsewhere at this stage. Should you do decide to do so, and we hope not, we would welcome a message to this effect.

As you know, EMBO Molecular Medicine has a "scooping protection" policy, whereby similar findings that are published by others during review or revision are not a criterion for rejection. However, I do ask you to get in touch with us after three months if you have not completed your revision, to update us on the status. Please also contact us as soon as possible if similar work is published elsewhere.

EMBO Molecular Medicine now requires a complete author checklist (<http://embomolmed.embopress.org/authorguide#editorial3>) to be submitted with all revised manuscripts. Provision of the author checklist is mandatory at revision stage; The checklist is designed to enhance and standardize reporting of key information in research papers and to support reanalysis and repetition of experiments by the community. The list covers key information for figure panels and captions and focuses on statistics, the reporting of reagents, animal models and human subject-derived data, as well as guidance to optimise data accessibility.

We now mandate that all corresponding authors list an ORCID digital identifier. You may acquire one through our web platform upon submission and the procedure takes <90 seconds to complete. We also encourage co-authors to supply an ORCID identifier, which will be linked to their name for unambiguous name identification.

Please carefully adhere to our guidelines for authors (<http://embomolmed.embopress.org/authorguide>) to accelerate manuscript processing in case of acceptance.

I look forward to receiving your revised manuscript in due time.

***** Reviewer's comments *****

Referee #1 (Comments on Novelty/Model System for Author):

It is unclear how the investigators show that lack of iBALT results in reduced emphysema; their expanded data shows no reduction in lung inflammation in mice lacking Ch25h or EB12.

Referee #1 (Remarks for Author):

The paper by Jia et al aims to identify the role of cholesterol metabolism in iBALT formation in CS-induced emphysema. The investigators have identified increased expression of CH25H and CYP7B1 in airway epithelia of smokers and mice exposed to chronic smoke. Mice deficient in CH25H or the oxysterol receptor EB12, exposed to smoke failed to form iBALT and were protected against emphysema. Systemic administration of clotrimazole resolved iBALT in the lungs and reversed CS-induced emphysema in mice.

Overall the study provides new information on association between CH25H in human COPD, and describes how iBALTs can form in the lungs of mice exposed to chronic CS. However, the study does not address the mechanisms for how iBALTs can cause emphysema. iBALTs are found in several chronic inflammatory diseases, and disrupting them has not altered disease pathogenesis in humans. iBALTs are highly associated with changes in airway microbiota, and treatment with clotrimazole, an anti-fungal antibiotic, could affect lung's local mycobiome. These findings suggest that development of organized secondary lymphoid structure in the lungs could be independent of COPD pathogenesis. Other areas that require some clarification are listed below including:

1- Figure 1: The authors show in panel D that CH25H is selectively expressed in emphysematous

regions within the lungs. What is the mechanism for the regional upregulation of CH25H in the lungs? Is this related to CS-exposure, or immune cell infiltration?

2- Figure 2: Ch25h^{-/-} mice are protected against CS-induced emphysema and iBALT formation. However, despite lack of iBALT, large number of activated macrophages, T and B cells are recruited into the lungs of these mice (shown in expanded data). These findings would imply that despite recruitment of immune cells mice are protected against emphysema. The mechanism for the protection against emphysema is unclear.

3- Did the authors measure whether neutrophils and or macrophages were activated in WT compared to Ch25h^{-/-} mice? Measurement of pro-inflammatory mediators and cytokines should be addressed in this model.

4- Figure 3: Consistent with data in Figure 1, the authors show Ebi2^{-/-} mice are resistant to CS-mediated emphysema; however, unlike Ch25h^{-/-} mice, they show reduced activated B cells (CD69/CD19). What is the mechanism for reduced B cell activation in Ebi2^{-/-} mice in this model? Supplementary expanded figure 2 shows in the absence of Ebi2 macrophage, neutrophils, T and B cells are recruited to the lungs. What cytokines/chemokines do these immune cells produce? Did the authors measure MMP12 or NE expression in the lungs of these mice?

Figure 5: The lung inflammatory cell profile remains largely intact in mice treated with clotrimazole. The authors should provide the mechanism for how clotrimazole can reverse emphysema.

Referee #2 (Comments on Novelty/Model System for Author):

Models and methodology of testing is well thought out and executed, with a high relevance to the disease studied. This study is a novel exploration of the mechanisms of pathogenesis in COPD, as well as exploring the area of immunometabolism and the molecular mechanisms underlying the development of iBALT. The study provides a useful insight into possible therapeutic targets and is an excellent starting point, but will require further refinement and testing before being applicable in a clinical setting. Nevertheless, it is an excellent proposal for a new therapy of medical importance.

Referee #2 (Remarks for Author):

Overall, a very good report investigating a novel mechanism of pathogenesis in COPD, the connection between metabolism and immune responses, and an exciting prospect for a new therapy in a challenging disease. Paper is well written, with meaningful and precise conclusions and robust methodology.

Major Suggestions

-In the first section of results (pg6), data is presented about the expression of CH25H and CYP7B1 in a number of human and mouse samples, but in some samples only data for CH25H is included. It would be good to see (or mention if there is no difference) the expression of both genes for the samples listed

-Methodology should include the technique for preparing CSE (number of cigarettes, method of preparation, etc.)

-Where possible, data from lung function should be included in all relevant mouse studies to determine if the protective effects of disrupting oxysterol metabolism result in functional improvements.

Minor suggestions

-bottom of pg 6, recommend changing "...CS-exposure and remained elevated for at least 16 weeks" to "...CS-exposure, and remained elevated throughout 16 weeks of smoke exposure."

-When discussing the expression of CH25H in isolated airway epithelial cells (pg 6), it would strengthen the argument to mention that this data is compared to healthy smoking controls, thereby demonstrating that this is a feature of the disease itself more than an effect of smoke exposure alone.

-In discussing the effects of clotrimazole, reference is made to its ability to "reverse experimental COPD" and "regenerate the lungs". While the evidence for a beneficial and protective effect is very compelling, caution should be used before suggesting an ability to reverse emphysematous damage over preventing additional damage. Given the nature and design of the experiment, it would be better worded to say that it attenuates disease rather than reversing the disease.

Referee #1 (Comments on Novelty/Model System for Author):

It is unclear how the investigators show that lack of iBALT results in reduced emphysema; their expanded data shows no reduction in lung inflammation in mice lacking Ch25h or EBi2.

To answer the reviewers concerns, we firstly apologize for the paucity in clarity about how a lack of iBALT results in reduced emphysema, and the apparent lack of reduction in lung inflammation from our expanded data. We have therefore returned to our original flow cytometry data of single cell suspensions from whole lung that had been lavaged, and examined more closely the recruitment of macrophages into the lung tissue of *Ch25h*^{-/-} mice. We previously only showed lymphocyte flow cytometry data, but now with further analysis of our data set in which we included a panel for macrophage and neutrophil markers. This clearly shows a reduction in the recruitment of macrophages specifically into the tissue, in contrast to the alveolar lumen (Fig EV2D-F). This data has been discussed in the results section page 8-9 “Diminished oxysterol pathways impaired iBALT formation and attenuated cigarette smoke-induced COPD” and new figures included in Fig EV2. Furthermore, we have previously shown that B cell deficient mice do not generate iBALT and that this prevented CS-induced emphysema by impairing the activation of macrophages and MMP12 upregulation (John-Schuster et al, 2014). We therefore have used qPCR analysis of lung homogenate to show an altered MMP12:Timpl ratio, in the lungs of *Ch25h*^{-/-} mice (Fig EV2H).

The inserted section on pages 8-9 reads “We have previously shown that B cell deficient mice do not generate iBALT and that this prevented CS-induced emphysema by impairing the activation of macrophages and MMP12 upregulation (John-Schuster et al, 2014). To address the mechanisms underlying the protection against emphysema in *Ch25h*^{-/-} mice, flow cytometric analysis was undertaken on the lavaged lungs to address the recruitment of macrophages into the lung tissue following CS-exposure. In contrast to the alveolar lumen, total F4/80⁺ macrophages and CD11c⁻ CD11b⁺ recruited macrophages were significantly reduced in the lungs of *Ch25h*^{-/-} mice compared to wild-type animals following chronic CS-exposure (Fig EV2D-F). In support, mRNA expression of *Adgre1* the gene for F4/80 (Fig EV2G) and the *Mmp12:Timpl* ratio (Fig EV2H) were significantly reduced in the lungs of *Ch25h*^{-/-} mice compared to wild-types, following exposure to CS. Furthermore, flow cytometric analysis revealed reduced Ly6g positive neutrophils in the lungs of *Ch25h*^{-/-} mice compared to wild-type animals following chronic CS-exposure (Fig EV2I-J).”

Referee #1 (Remarks for Author):

The paper by Jia et al aims to identify the role of cholesterol metabolism in iBALT formation in CS-induced emphysema. The investigators have identified increased expression of CH25H and CYP7B1 in airway epithelia of smokers and mice exposed to chronic smoke. Mice deficient in CH25H or the oxysterol receptor EB12, exposed to smoke failed to form iBALT and were protected against emphysema. Systemic administration of clotrimazole resolved iBALT in the lungs and reversed CS-induced emphysema in mice.

Overall the study provides new information on association between CH25H in human COPD, and describes how iBALTs can form in the lungs of mice exposed to chronic CS. However, the study does not address the mechanisms for how iBALTs can cause emphysema. iBALTs are found in several chronic inflammatory diseases, and disrupting them has not altered disease pathogenesis in humans. iBALTs are highly associated with changes in airway microbiota, and treatment with clotrimazole, an anti-fungal antibiotic, could affect lung's local mycobiome. These findings suggest that development of organized secondary lymphoid structure in the lungs could be independent of COPD pathogenesis. Other areas that require some clarification are listed below including:

To address the effect of the anti-fungal drug clotrimazole on the lung mycobiome we collaborated with our colleagues at the Helmholtz Zentrum München who have extensive experience in analyzing the microbiome. Firstly, I would like to highlight that clotrimazole is an antifungal agent, developed to inhibit the cytochrome P450-dependent lanosterol 14- α -demethylase (Maertens, 2004), which accounts for its ability to inhibit mammalian cytochrome P450- enzymes like CYP7B1. We therefore assessed the affect of clotrimazole solely upon the lungs mycobiome, as suggested.

To this end we analyzed DNA extracted from snap frozen whole lung for the presence of fungal DNA by qPCR using the following Method:

Frozen lung powder was resuspended in 200 ul ultra pure water, 100 uL were retained as a backup. DNA was extracted using a combination of bead beating (6.0 m/s; 30 seconds) in phenol chloroform Isoamyl alcohol and the PureLink genomic DNA kit (ThermoFisher scientific). In addition, we tested the extraction protocol for suitability by extracting isolates from an *Aspergillus* strain (*A. Clavatus* 107 (DSM816)) and a yeast (*Candida cylindracea* ms-5 (DSM2031)). The cleanliness of the DNA was analyzed with a Nanodrop and the exact amount of DNA measured with the Fragment analyzer Genomic DNA 50kb Analysis kit.

The qPCR was carried out with an established universal fungus-specific ITS1/ITS4 primer pair (White et al, 1990). None of the extracts showed an amplicon (with very good running efficiency of 96.5% or 102.5% and 40 cycles). We therefore performed an additional PCR with the primer combination ITS86F/ITS4, a targeted region of DNA found widely across the Pilzphyla, and has been recommended several times for amplicon sequencing (Ihrmark et al, 2012; Op De Beeck et al, 2014; Vancov & Keen, 2009). Again, there was no amplification or one that goes beyond the greatest dilution of the standard (~ 1 copy number/ng DNA) (no longer distinguishable from a non-template control).

As can be observed (Table 1 below), we cannot detect any fungal DNA in the lungs of our mice, and can therefore only conclude that no effect of clotrimazole can be evidenced. Within the limits of detection, the effect of clotrimazole in reducing iBALT following cigarette smoke exposure therefore manifests from its ability to impair the enzyme CYP7B1 and the reduced generation of oxysterols. We have chosen not to include this data in the manuscript but leave it only in our response here. We trust that we have addressed your concerns by undertaking this further analysis. A summary of these results can be found below in Table 1.

Group	Number of lungs analyzed	ITS1xITS4 (40 cycles)	ITS86xITS4 (40 cycles)
FA 4m	4	No amplification	<1 copy/ng DNA
CS 4m	4	No amplification	<1 copy/ng DNA
FA 4m + CLT 2m	4	No amplification	<1 copy/ng DNA
CS 4m + CLT 2m	4	No amplification	<1 copy/ng DNA
FA 6m	4	No amplification	<1 copy/ng DNA
CS 6m	4	No amplification	<1 copy/ng DNA
FA 6m + CLT 2m	2	No amplification	<1 copy/ng DNA
CS 6m + CLT 2m	3	No amplification	<1 copy/ng DNA

Table 1. Fungal specific qPCR undertaken to assess fungal microbiome of the lung.

1- Figure 1: The authors show in panel D that CH25H is selectively expressed in emphysematous regions within the lungs. What is the mechanism for the regional upregulation of CH25H in the lungs? Is this related to CS-exposure, or immune cell infiltration?

Response 1: Thank you very much for highlighting such a pertinent point. To address this avenue of investigation we have undertaken a series of new experiments for the manuscript. That stems from the observation that CH25H upregulation is driven by TLR4 Myd88 independent signaling (Diczfalusy et al, 2009). We now demonstrate that stimulating two human airway epithelial cell lines (BEAS-2B and HBE) with the TLR4 agonist LPS, induces CH25H expression comparable to cigarette smoke extract (Fig EV1D and E). Additionally, we show that the greatest level of TLR4 expression in COPD patients is in the epithelial cells of airways associated with emphysematous tissue (Fig EV1C), and using gene set enrichment analysis (GSEA, Broad Institute) that TLR4 signalling pathways are enriched in the airways of COPD patients compared to healthy smoker controls (Fig EV1B). Furthermore, in addition to cigarette smoke directly acting upon the airway epithelial cells to induce CH25H upregulation, we show that TNF- α can also induce expression (Fig

EV1F), suggesting that immune cell infiltration and cytokine release may also enhance the expression of *CH25H*. This data has been inserted into Fig. EV1B-E, and discussed in the results section “Oxysterol metabolism increases in airway epithelial cells of COPD patients and mouse” page 7.

The inserted section reads “To address the mechanism underlying the regional upregulation of *CH25H* predominantly localized to the airways in COPD patients and in particular that associated with emphysematous tissue, we first undertook gene set enrichment analysis (GSEA) (Mootha et al, 2003; Subramanian et al, 2005) on the publically available transcriptomics dataset of small airway epithelial cells from COPD patients described above (Tilley et al, 2011). *CH25H* upregulation is driven by TLR4 Myd88 independent signaling (Diczfalusy et al, 2009), indeed GSEA revealed a strong enrichment of both total TLR and TLR4 dependent signaling in small airway epithelial cells taken from the lungs of COPD patients compared to smoking controls (Fig. EV1B). Supporting a recent observation that *TLR4* expression is increased in the airways of COPD patients (Haw et al, 2017). Furthermore, staining of airway sections revealed a strong increase in TLR4 expression localized to the airways of emphysematous COPD patients rather than non-emphysematous or healthy control airways (Fig. EV1C). Additionally, treating human bronchial epithelial cell lines with the TLR4 agonist LPS, induced expression of *CH25H* similar to that observed with cigarette smoke (Fig. EV1D and E). Interestingly, the pro-inflammatory cytokine TNF- α alone was also able to induce enhanced *CH25H* expression in airway epithelial cells, suggesting that the pro-inflammatory environment in addition to direct effects of CS-exposure upon the airway epithelial cells is capable of enhancing *CH25H* expression.”

2- Figure 2: Ch25h^{-/-} mice are protected against CS-induced emphysema and iBALT formation. However, despite lack of iBALT, large number of activated macrophages, T and B cells are recruited into the lungs of these mice (shown in expanded data). These findings would imply that despite recruitment of immune cells mice are protected against emphysema. The mechanism for the protection against emphysema is unclear.

Response 2: Similar to our response raised to the first point above discussing how lack of iBALT results in reduced emphysema, we are now convinced that our more thorough analysis of the recruitment of cells into the lung and the differences highlighted between wild-type mice and *Ch25h^{-/-}* mice go some way in answering this point. This further analysis of our flow cytometric data sets included a panel of macrophage and neutrophil markers. This clearly shows a reduction in the recruitment of macrophages specifically into the tissue, in contrast to the alveolar lumen (Fig EV2D-F). In support, mRNA expression of *Adgre1* the gene for F4/80 was significantly reduced in the lungs of *Ch25h^{-/-}* mice compared to wild-type, following exposure to chronic CS (Fig EV2G). Our flow cytometric analysis also revealed reduced Ly6g positive neutrophils in the lungs of *Ch25h^{-/-}* mice compared to wild-type animals following chronic CS-exposure (Fig EV2I-J). Furthermore, we have previously shown that B cell deficient mice do not generate iBALT and that this prevented CS-induced emphysema by impairing the activation of macrophages and MMP12 upregulation (John-Schuster et al, 2014). We therefore have used qPCR analysis of lung homogenate to show an altered MMP12:Timpl ratio (Fig EV2H), in the lungs of *Ch25h^{-/-}* mice. This data has been discussed in the results section page 8-9 “Diminished oxysterol pathways impaired iBALT formation and attenuated cigarette smoke-induced COPD”.

3- Did the authors measure whether neutrophils and or macrophages were activated in WT compared to Ch25h^{-/-} mice? Measurement of pro-inflammatory mediators and cytokines should be addressed in this model.

Response 3: We would kindly refer the reviewer to the heat map of chemokine and cytokine expression presented in Fig. EV2B and discussed on page 8. We analyzed the expression of, in addition to oxysterol metabolizing enzymes, the chemokines Cxcl13, Cxcl9, Ccl19 and Ccl21 chemoattractants for B cells, T cells and DCs, and show equivalent expression between WT and *Ch25h^{-/-}* mice following chronic CS exposure. We would like to re-iterate in support of this that we do not show impaired recruitment of lymphocytes to the lung (Fig 2E), neither do we wish to argue that to be the case, but rather in the absence of oxysterols the lymphocytes that migrate into the lung cannot arrange into iBALT structures, as we demonstrated (Fig 2B-D). Our heat map also analyzed expression of CXCL1, GM-CSF, MCP1, TNF- α and IL-1 α , important inflammatory cytokines. In addition our new data analysis (Fig EV2H) and page 8 clearly demonstrated a reduced

MMP12:TIMP1 ratio in the lungs of *Ch25h*^{-/-} mice compared to wild-type animals following exposure to chronic CS.

4- Figure 3: Consistent with data in Figure 1, the authors show *Ebi2*^{-/-} mice are resistant to CS-mediated emphysema; however, unlike *Ch25h*^{-/-} mice, they show reduced activated B cells (CD69/CD19). What is the mechanism for reduced B cell activation in *Ebi2*^{-/-} mice in this model? Supplementary expanded figure 2 shows in the absence of *Ebi2* macrophage, neutrophils, T and B cells are recruited to the lungs. What cytokines/chemokines do these immune cells produce? Did the authors measure MMP12 or NE expression in the lungs of these mice?

Response 4: This is a really noteworthy point raised by the reviewer therefore we have undertaken new experiments on splenic B cells isolated from WT and *Ebi2*^{-/-} mice to address this point. Firstly, we confirm as in our *in vivo* results that *ex vivo* activated B cell cells from *Ebi2*-deficient mice, by BCR cross-linking, also show impaired activation demonstrated as reduced CD69 upregulation (Fig EV3C). CD69 expression is regulated by the primary response gene *Egr1* (Richards et al, 2001; Vazquez et al, 2009), which interestingly has impaired upregulation in *Ebi2*^{-/-} B cells compared to wild-type cells (Fig EV3E). The new data is presented in Fig. EV3C-E, and discussed in the section “Diminished oxysterol pathways impaired iBALT formation and attenuated cigarette smoke-induced COPD” page 9.

The inserted section on page 9 reads “To address this, splenic B cells were isolated from *Ebi2*^{-/-} and wild-type mice and activated *ex vivo* by IgM cross-linking. Similar to the *in vivo* situation flow cytometric analysis revealed reduced activation of *Ebi2*^{-/-} B cells as demonstrated by less upregulation of the surface activation marker CD69 (Fig EV3C), which was accompanied by reduced MHC II expression (Fig EV3D). CD69 expression in B cells is regulated by *Egr1* (Richards et al, 2001; Vazquez et al, 2009), a primary response gene rapidly induced in B cells following BCR cross-linking (McMahon & Monroe, 1995; Seyfert et al, 1989). Interestingly, *Ebi2*^{-/-} B cells 6h post BCR cross-linking upregulated *Egr1* less than wild-type B cells (Fig EV3E), proposing that the impaired activation of *Ebi2*^{-/-} B cells may stem from an inability to fully induce expression of the early response gene *Egr1*. Future work should determine further the role of *Ebi2* in *Egr1* transcriptional regulation.”

To answer the point referring to the analysis of cytokine and chemokine expression levels in the *Ebi2*^{-/-} mice, we have now undertaken qPCR analysis of total lung homogenate for all genes analyzed in the *Ch25h*^{-/-} mice in *Ebi2*^{-/-} animals and generated a heat map of this data. This new data is presented in Fig EV3B and referred to on page 9. This also included expression of the oxysterol metabolizing enzymes. As with the *Ch25h*-deficient animals there was no difference in the expression of the chemokines *Cxcl13*, *Cxcl9*, *Ccl19* and *Ccl21* between WT and *Ebi2*^{-/-} mice following chronic CS exposure. To address macrophages we assessed the level of *Mcp1* and *Tnf-α*, critical for macrophage recruitment and a key cytokine predominantly secreted by activated macrophages, respectively. GM-CSF levels were also analyzed due to its ability to activate both neutrophils and macrophages, with *Cxcl1* (KC) levels analysed because of its neutrophil chemotactic ability. The following text was inserted into the results section on page 9 “Diminished oxysterol pathways impaired iBALT formation and attenuated cigarette smoke-induced COPD” stating: “with cytokine and chemokine expression profiles similar to that observed for *Ch25h*^{-/-} and wild-type mice (Fig EV3B)” to highlight this data.

5- Figure 5: The lung inflammatory cell profile remains largely intact in mice treated with clotrimazole. The authors should provide the mechanism for how clotrimazole can reverse emphysema.

Response 5: Our previous data (John-Schuster et al, 2014) and that presented in this paper, clearly identifies the generation of iBALT as necessary to the development of CS-induced emphysema. Besides B cell deficient mice, another two independent genetically modified mice strains *Ch25h* and *Ebi2*, all fail to develop iBALT and are protected against emphysema development driven by cigarette smoke (Fig. 2 and 3). In the elastase model of emphysema in which iBALT does not form, these mice strains are not protected (Fig EV6 in this manuscript). Pharmacological inhibition of oxysterol synthesis using clotrimazole, in our *ex vivo* studies presented in this manuscript (Fig. 4E-G), clearly demonstrate that oxysterol generation is inhibited in murine airways exposed to cigarette smoke extract and treated with clotrimazole, and that this prevents the migration of B cells to the airway. Similar to the lack of B cell migration observed towards CS-exposed airways deficient in the oxysterol metabolizing enzyme *CH25H* (Fig. 4A-D). We therefore propose that *in vivo* treatment

of mice with clotrimazole inhibits cigarette smoke induced oxysterol production by the airway epithelium, which disrupts the organization of lymphocytes into functional iBALT structures upon the airway, and subsequent attenuation of emphysema. We have previously shown that in B cell deficient mice which do not generate iBALT, this prevented CS-induced emphysema by impairing the activation of macrophages and MMP12 upregulation (John-Schuster et al, 2014). Indeed, in the lavage of the clotrimazole treated mice exposed to cigarette smoke there is reduced macrophage infiltration (Fig. 5E). In line with this, inhalation of alendronate which reduced macrophage accumulation following 6 months of CS exposure prevented emphysematous changes (Ueno et al, 2015). Future work should determine if aerosilization of clotrimazole for local drug delivery to the airways is also sufficient to impair oxysterol synthesis and attenuate COPD disease progression by disrupting iBALT generation, similar to what we observed following systemic treatment, and thus offer a viable therapeutic alternative.

Referee #2 (Comments on Novelty/Model System for Author):

Models and methodology of testing is well thought out and executed, with a high relevance to the disease studied. This study is a novel exploration of the mechanisms of pathogenesis in COPD, as well as exploring the area of immunometabolism and the molecular mechanisms underlying the development of iBALT. The study provides a useful insight into possible therapeutic targets and is an excellent starting point, but will require further refinement and testing before being applicable in a clinical setting. Nevertheless, it is an excellent proposal for a new therapy of medical importance.

Referee #2 (Remarks for Author):

Overall, a very good report investigating a novel mechanism of pathogenesis in COPD, the connection between metabolism and immune responses, and an exciting prospect for a new therapy in a challenging disease. Paper is well written, with meaningful and precise conclusions and robust methodology.

Major Suggestions

-In the first section of results (pg6), data is presented about the expression of CH25H and CYP7B1 in a number of human and mouse samples, but in some samples only data for CH25H is included. It would be good to see (or mention if there is no difference) the expression of both genes for the samples listed

Response: Thank you very much for the question. We have included below the data you request. As you can see (Fig. 1 below) there is no significant change to the expression of *CYP7B1* in emphysematous versus non emphysematous regions of COPD lung tissue. Additionally, *Cyp7b1* expression in micro-dissected murine airways after *in vivo* CS-exposure did not vary over time.

Fig 1. A: *CYP7B1* mRNA abundance in COPD patient lung core biopsy samples taken from areas of non-emphysematous and emphysematous tissue. B: *Cyp7b1* mRNA abundance in isolated airways from C57BL/6 mice exposed to cigarette smoke (CS) for the duration indicated, shown relative to filtered air (FA), one experiment with five mice per group.

-Methodology should include the technique for preparing CSE (number of cigarettes, method of preparation, etc.)

Response: We apologise for our oversight, and have now included this section in the Materials and Methods section of the manuscript. We have inserted the following text on page 24-25: "Cigarette smoke extract (CSE) preparation. CSE was generated by bubbling smoke from three research cigarettes (3R4F, Tobacco Research Institute, University of Kentucky) through 30ml of airway epithelial cell culture medium (PromoCell) at a puffing speed equating to one cigarette every 5 mins, in a closed environment with limited air flow. This solution was taken as 100% CSE.

-Where possible, data from lung function should be included in all relevant mouse studies to determine if the protective effects of disrupting oxysterol metabolism result in functional improvements.

Response: As a group we feel that stereological analysis of lung tissue to be a much more sensitive tool in analyzing the changes occurring in the lung during the development of CS-induced COPD in the mouse model. However, we take on board the wishes of the reviewer to analyse this data, and to that end have included lung function data in Fig 2 below. The protective effects of disrupting oxysterol metabolism and its impairment upon iBALT formation and emphysema development as determined by stereological analysis, can also be detected as functional improvements in lung function in the *Ebi2*-deficient mice and those treated with clotrimazole to four months.

Fig 2. Lung function as lung compliance normalized to body weight and total lung capacity (TLC) was determined using a forced pulmonary maneuver system (Buxco Research Company, Data Sciences International) running FinePointe Software in WT mice and *Ch25h*^{-/-} (A), *Ebi2*^{-/-} (B), and clotrimazole treated mice (C) exposed to chronic cigarette smoke for the duration indicated. Data shown are mean \pm s.d. , * p <0.05, ** p <0.01, *** p <0.001 and **** p <0.0001 one-way ANOVA and Turkey's multiple comparisons test.

Minor suggestions

-bottom of pg 6, recommend changing "...CS-exposure and remained elevated for at least 16 weeks" to "...CS-exposure, and remained elevated throughout 16 weeks of smoke exposure."

Response: We have corrected the text as suggested, thank you.

-When discussing the expression of *CH25H* in isolated airway epithelial cells (pg 6), it would strengthen the argument to mention that this data is compared to healthy smoking controls, thereby demonstrating that this is a feature of the disease itself more than an effect of smoke exposure alone.

Response: Thank you very much for that insightful comment. That is a very important point that indeed strengthens the argument and as such we have taken on board your suggestion and amended the text on page 6 which now reads: "*CH25H* mRNA expression was elevated in isolated airway epithelial cells from COPD patients compared to healthy smoking controls (fourth independent cohort) (Fig 1F), as...".

-In discussing the effects of clotrimazole, reference is made to its ability to "reverse experimental COPD" and "regenerate the lungs". While the evidence for a beneficial and protective effect is very compelling, caution should be used before suggesting an ability to reverse emphysematous damage

over preventing additional damage. Given the nature and design of the experiment, it would be better worded to say that it attenuates disease rather than reversing the disease.

Response: We take on board your point, and agree we may have been a little hasty and caught up in the excitement of our data. To that end we have amended our manuscript, so that we state that we have attenuated disease rather than reversed and regenerated lungs.

References

- Diczfalusy U, Olofsson KE, Carlsson AM, Gong M, Golenbock DT, Rooyackers O, Flaring U, Bjorkbacka H (2009) Marked upregulation of cholesterol 25-hydroxylase expression by lipopolysaccharide. *J Lipid Res* 50: 2258-2264
- Haw TJ, Starkey MR, Pavlidis S, Fricker M, Arthurs AL, Mono Nair P, Liu G, Hanish I, Kim RY, Foster PS, Horvat JC, Adcock IM, Hansbro PM (2017) Toll-like receptor 2 and 4 have Opposing Roles in the Pathogenesis of Cigarette Smoke-induced Chronic Obstructive Pulmonary Disease. *Am J Physiol Lung Cell Mol Physiol*: ajplung 00154 02017
- Ihrmark K, Bodeker IT, Cruz-Martinez K, Friberg H, Kubartova A, Schenck J, Strid Y, Stenlid J, Brandstrom-Durling M, Clemmensen KE, Lindahl BD (2012) New primers to amplify the fungal ITS2 region--evaluation by 454-sequencing of artificial and natural communities. *FEMS Microbiol Ecol* 82: 666-677
- John-Schuster G, Hager K, Conlon TM, Irmeler M, Beckers J, Eickelberg O, Yildirim AO (2014) Cigarette smoke-induced iBALT mediates macrophage activation in a B cell-dependent manner in COPD. *Am J Physiol Lung Cell Mol Physiol* 307: L692-706
- Maertens JA (2004) History of the development of azole derivatives. *Clin Microbiol Infect* 10 Suppl 1: 1-10
- McMahon SB, Monroe JG (1995) Activation of the p21ras pathway couples antigen receptor stimulation to induction of the primary response gene *egr-1* in B lymphocytes. *J Exp Med* 181: 417-422
- Mootha VK, Lindgren CM, Eriksson KF, Subramanian A, Sihag S, Lehar J, Puigserver P, Carlsson E, Ridderstrale M, Laurila E, Houstis N, Daly MJ, Patterson N, Mesirov JP, Golub TR, Tamayo P, Spiegelman B, Lander ES, Hirschhorn JN, Altshuler D, Groop LC (2003) PGC-1alpha-responsive genes involved in oxidative phosphorylation are coordinately downregulated in human diabetes. *Nat Genet* 34: 267-273
- Op De Beeck M, Lievens B, Busschaert P, Declerck S, Vangronsveld J, Colpaert JV (2014) Comparison and validation of some ITS primer pairs useful for fungal metabarcoding studies. *PLoS One* 9: e97629
- Richards JD, Dave SH, Chou CH, Mamchak AA, DeFranco AL (2001) Inhibition of the MEK/ERK signaling pathway blocks a subset of B cell responses to antigen. *J Immunol* 166: 3855-3864
- Seyfert VL, Sukhatme VP, Monroe JG (1989) Differential expression of a zinc finger-encoding gene in response to positive versus negative signaling through receptor immunoglobulin in murine B lymphocytes. *Mol Cell Biol* 9: 2083-2088
- Subramanian A, Tamayo P, Mootha VK, Mukherjee S, Ebert BL, Gillette MA, Paulovich A, Pomeroy SL, Golub TR, Lander ES, Mesirov JP (2005) Gene set enrichment analysis: a knowledge-based approach for interpreting genome-wide expression profiles. *Proc Natl Acad Sci U S A* 102: 15545-15550
- Tilley AE, O'Connor TP, Hackett NR, Strulovici-Barel Y, Salit J, Amoroso N, Zhou XK, Raman T, Omberg L, Clark A, Mezey J, Crystal RG (2011) Biologic phenotyping of the human small airway epithelial response to cigarette smoking. *PLoS One* 6: e22798

Ueno M, Maeno T, Nishimura S, Ogata F, Masubuchi H, Hara K, Yamaguchi K, Aoki F, Suga T, Nagai R, Kurabayashi M (2015) Alendronate inhalation ameliorates elastase-induced pulmonary emphysema in mice by induction of apoptosis of alveolar macrophages. *Nat Commun* 6: 6332

Vancov T, Keen B (2009) Amplification of soil fungal community DNA using the ITS86F and ITS4 primers. *FEMS Microbiol Lett* 296: 91-96

Vazquez BN, Laguna T, Carabana J, Krangel MS, Lauzurica P (2009) CD69 gene is differentially regulated in T and B cells by evolutionarily conserved promoter-distal elements. *J Immunol* 183: 6513-6521

White TJ, Bruns TD, Lee SB, Taylor JW (1990) Amplification and direct sequencing of fungal ribosomal RNA Genes for phylogenetics. In *PCR - Protocols and Applications - A Laboratory Manual.*, Innis MA, Gelfand DH, Sninsky JJ, White TJ (eds), pp 315-322. Cambridge: Academic Press

2nd Editorial Decision

9 February 2018

Thank you for the submission of your manuscript to EMBO Molecular Medicine. We have now heard back from the two referees whom we asked to re-evaluate your manuscript.

You will see that while referee 2 is now satisfied, referee 1 is not and still raises important concerns, mainly that the iBALT formation being necessary and sufficient for smoke-induced emphysema development is not fully supported by the data.

I would like to give you another chance to address this concern. This is an unusual practice for us as we normally support a single round of revision; I can't stress enough that this would be the last chance to convince this referee. Would you be amenable to this? Could you please let me know in a few lines what your plans would be?

***** Reviewer's comments *****

Referee #1 (Remarks for Author):

The revised paper has addressed several of the previous concerns regarding the mechanism of ch25h in CS-exposed emphysema. The authors states that they further analyzed ch25h^{-/-} mice flow data and found despite intact activated B, and T cells (Fig 2) in the lungs, macrophage and neutrophils infiltration in the lung but not BAL of ch25h null smoke exposed mice are reduced (Extended figure 2 panels C and E-H). They believe reduction of lung macrophage protects these mice against smoke induced emphysema.

My remaining concern is the authors' premise that iBALT formation is necessary and sufficient for recruitment and activation of macrophages in the lungs in response to cigarette smoke. Several prior studies have shown that SCID and Rag^{-/-} mice that lack B and T cells are not protected from mainstream smoke induced emphysema (D'Hulst et al *Respir Res.* 2005). Furthermore, it is unclear how reduced macrophages recruitment is confined to the lungs but not in the BAL?

Referee #2 (Comments on Novelty/Model System for Author):

Experiments have been well designed and performed, and address the research hypothesis in a robust manner. Authors have addressed all reviewer concerns appropriately.

Authors' response

20 February 2018

We are emailing you in your capacity as Senior Editor of EMBO Molecular Medicine, and would like to thank you in advance for your time handling our manuscript referenced above.

Thank you for enabling us the opportunity to address the concerns one referee voices over part of our manuscript. We strongly believe that we have an exciting and highly citable set of data that would be very much of interest to a wide scientific audience, because of the prospects that blocking the oxysterol pathway of cholesterol metabolism holds for disrupting tertiary lymphoid organs in addition to iBALT, and the plethora of pathologies driven by this lymphoid tissue, including autoimmunity and transplant rejection.

You request a response outlining our plans given the review of the first revision of our submission. We strongly believe that we can refute this referee's concern, by using a combination of existing literature and data we have generated. Although we are highly appreciative of the reviewer's concerns, we respectfully yet strongly disagree with statements provided by this reviewer. In particular, the notion that SCID and Rag^{-/-} mice (that lack B and T cells) are not protected from mainstream smoke-induced emphysema, referenced to the publication from D'Hulst et al. 2005 in the journal *Respiratory Research* (Vol. 6 pg. 147, (IF 3,8)), is incorrect. We would like to politely highlight that the above-mentioned article only refers to SCID mice and not Rag mice, although the referee inferred that both strains were examined in this publication. Additionally, the SCID mice were on a Balb/c background, as opposed to more recent publications from that group, as well as ours and others, which largely used mice on the B6 background.

After a six month exposure to cigarette smoke (CS), SCID mice demonstrated equal development of emphysema, however, the number of macrophages and neutrophils were significantly increased in SCID mice compared with wild-type mice exposed to CS (Fig 1 from that paper). This would suggest that i) the enhanced innate response is developmentally compensating for the lack of lymphocytes in the SCID mouse, ii) this process highlights the role of macrophages in driving emphysema development, which is further enhanced because of the stronger innate immune response in Balb/c (the background of the SCID mice used) compared with the B6 strain after CS exposure (Botelho et al. 2010 *American journal of respiratory cell and molecular biology* 42:394).

Furthermore, we examined the literature for studies that examined the effect of CS-induced emphysema in Rag knock-out mice. We found an article that assessed only CS-induced inflammation in Rag knock-out mice (Botelho et al. 2010 *American journal of respiratory cell and molecular biology* 42:394). Interestingly, this publication additionally compared the inflammatory response between mouse strains. The authors clearly showed that lung inflammation after 5 weeks of CS exposure was significantly higher in Rag1 knock-out mice than in wild type animals, again indicative of an enhanced innate response in animals lacking lymphocytes from birth. The authors themselves discuss that this increase in innate cells may be compensating for the developmentally lack of lymphocytes.

In contradiction to early findings from the group of Prof. Guy Brusselle (publication from D'Hulst et al. 2005), two recent articles from the same group (Bracke et al. 2013 *American journal of respiratory and critical care medicine* 188:343, cited in our manuscript, and Seys et al. 2015 *American journal of respiratory and critical care medicine* 192: 706) investigated the role of iBALT in emphysema. Here, treatment of mice exposed to chronic cigarette smoke, both prophylactically and therapeutically, with an anti-CXCL13 antibody prevented CS-induced iBALT formation (Bracke et al. 2013, Fig. 4) and prophylactically this prevented alveolar wall destruction (same paper, Fig. 9). In support of this, subsequent work by the same group (Seys et al. 2015) used a BAFF-receptor fusion protein to target iBALT formation. The authors demonstrated a reduction of B cell and interstitial macrophage numbers in the lungs of BAF-targeted CS-exposed mice (Fig. 8), which were accompanied by reduced numbers of CS-induced iBALT structures (Fig. 4). Prophylactic treatment with the fusion protein also attenuated the CS-induced destruction of alveolar walls (Fig. 9).

Finally, we would like to highlight the growing body of evidence that iBALT size and number correlates with the severity of COPD in patients. Prominent work was published two years ago analyzing transcriptomics data of emphysematous COPD patients versus those only presenting bronchiolitis, which clearly demonstrated a distinct B cell signature in the emphysematous patients (Faner et al. 2016 *American journal of respiratory and critical care medicine* 193:1242). Additionally, recent work by Polverino et al. (2015 *American journal of respiratory and critical care medicine* 192:695) elegantly demonstrated that patients with severe COPD (GOLD stages III/IV,

mostly with emphysema) had more numerous and larger iBALT structures than more mild patients (GOLD stages I/II).

In summary, we therefore strongly disagree with the bias voiced by one reviewer, citing one publication as opposed to the overwhelming evidence of a number of more recent publications that support our data. We do feel strongly that this bias is voiced unfairly, and would point out that addressing this criticism experimentally would take a minimum of 12 months to be fully addressable (exposing SCID or Rag knock out mice on the B6 background to chronic cigarette smoke, which will include new breedings and analysis).

2nd Revision - authors' response

22 February 2018

Referee #1 (Remarks for Author):

The revised paper has addressed several of the previous concerns regarding the mechanism of ch25h in CS-exposed emphysema. The authors states that they further analyzed ch25h-/- mice flow data and found despite intact activated B, and T cells (Fig 2) in the lungs, macrophage and neutrophils infiltration in the lung but not BAL of ch25h null smoke exposed mice are reduced (Extended figure 2 panels C and E-H. They believe reduction of lung macrophage protects these mice against smoke induced emphysema.

My remaining concern is the authors' premise that iBALT formation is necessary and sufficient for recruitment and activation of macrophages in the lungs in response to cigarette smoke. Several prior studies have shown that SCID and Rag-/- mice that lack B and T cells are not protected from mainstream smoke induced emphysema (D'Hulst et al Respir Res. 2005).

Furthermore, it is unclear how reduced macrophages recruitment is confined to the lungs but not in the BAL?

Although we are highly appreciative of the reviewer's concerns, we respectfully yet strongly disagree with statements provided by this reviewer. In particular, the notion that SCID and Rag-/- mice (that lack B and T cells) are not protected from mainstream smoke-induced emphysema, referenced to the publication from D'Hulst et al. 2005 in the journal Respiratory Research (1), is incorrect. We would like to politely highlight that the above-mentioned article only refers to SCID mice and not Rag mice, although the referee inferred that both strains were examined in this publication. Additionally, the SCID mice were on a Balb/c background, as opposed to more recent publications from that group, as well as ours and others, which largely used mice on the B6 background.

After a six month exposure to cigarette smoke (CS), SCID mice demonstrated equal development of emphysema, however, the number of macrophages and neutrophils were significantly increased in SCID mice compared with wild-type mice exposed to CS (Fig 1 from (1)). This would suggest that i) the enhanced innate response is developmentally compensating for the lack of lymphocytes in the SCID mouse, ii) this process highlights the role of macrophages in driving emphysema development, which is further enhanced because of the stronger innate immune response in Balb/c (the background of the SCID mice used) compared with the B6 strain after CS exposure (2). The authors of this paper themselves discuss that this increase in innate cells may be compensating for the developmental lack of lymphocytes.

Furthermore, we examined the literature for studies that examined the effect of CS-induced emphysema in Rag knock-out mice. We found an article that assessed only CS-induced inflammation in Rag knock-out mice (2). Interestingly, this publication additionally compared the inflammatory response between mouse strains. The authors clearly showed that lung inflammation after 5 weeks of CS exposure was significantly higher in Rag1 knock-out mice than in wild type animals, again indicative of an enhanced innate response in animals lacking lymphocytes from birth.

In contradiction to the D'Hulst et al. 2005 publication of Prof. Guy Brusselle (1), two recent articles in the American journal of respiratory and critical care medicine from the same group (3) cited in our manuscript and (4), now both of them cited in our manuscript, investigated the role of iBALT in

emphysema. Here, treatment of mice exposed to chronic cigarette smoke, both prophylactically and therapeutically, with an anti-CXCL13 antibody prevented CS-induced iBALT formation (Fig. 4 from (3)) and prophylactically this prevented alveolar wall destruction (Fig. 9 from (3)). In support of this, subsequent work by the same group (4) used a BAFF-receptor fusion protein to target iBALT formation. The authors demonstrated a reduction of B cell and interstitial macrophage numbers in the lungs of BAFF-targeted CS-exposed mice (Fig. 8 from (4)), which were accompanied by reduced numbers of CS-induced iBALT structures (Fig. 4 from (4)). Prophylactic treatment with the fusion protein also attenuated the CS-induced destruction of alveolar walls (Fig. 9 from (4)).

The innate immune response is clearly important to the development of CS-induced emphysema (5, 6), which is why we believe it was important to highlight that in the lungs of our *Ch25h*^{-/-} mice after CS exposure, that lacked iBALT formation, there was a reduction in both recruited macrophages and neutrophils in the lungs of these mice as demonstrated by flow cytometry and gene expression (Manuscript Fig. EV2D-J, now Fig. EV2D-K). To substantiate this, we have included a figure below (Fig. 1) that is of immunohistochemical analysis for Galectin-3, which shows an increase of macrophages in the lungs of wild type mice after CS exposure that is not present in knock-out animals.

This has also been inserted into our manuscript as Fig. EV2G, and an insertion into the final sentence of the results section on page 8 which reads: “In support, immunohistochemically stained galectin-3 positive macrophages (Fig EV2G), mRNA expression of *Adgre1* the gene for F4/80 (Fig EV2H) and the *Mmp12:Timp1* ratio (Fig EV2I) were significantly reduced in the lungs of *Ch25h*^{-/-} mice compared to wild-type, following exposure to CS.” The figure legend and Methods have also been updated.

Of interest, one can observe an accumulation of macrophages around the iBALT structures. This was something we also observed in a prior publication (7), where we showed that mice deficient in B cells did not generate iBALT structures after CS exposure and were protected against emphysema development. This is again data highlighting that B cells and iBALT formation are important for the recruitment of macrophages as instigators of emphysema following CS exposure. Furthermore, in our prior publication, as well as demonstrating that the macrophage number was reduced in the lungs of mice that lacked iBALT, we additionally included an in vitro experiment showing that B cell-derived IL-10 drives macrophage activation and MMP12 upregulation (7), which was discussed in our manuscript on page 8. To support and substantiate the novel concept that B cells can regulate macrophage function and recruitment in COPD, we would like to refer to two key publications. Firstly, in 2012 it was elegantly demonstrated in a model of viral infection that B cells were crucial for the maintenance of subcapsular sinus macrophages of murine lymph nodes (8). The following year B cells were demonstrated to trigger Ly6c^{hi} monocyte mobilization, the same population that leads to the generation of recruited macrophages in the lung, and impair heart function after acute myocardial infarction (9).

Fig. 1 Representative immunohistochemical images of wild type (WT) and CH25H-deficient (*Ch25h*^{-/-}) mice exposed to filtered air (FA) or cigarette smoke (CS) for 4 months, stained to detect galectin-3 (Red) and hematoxylin counter stained. Revealing a greater influx of macrophages in the wild type mice exposed to CS, which cluster around the iBALT structures. Scale bar 100 μ m.

In terms of reduced macrophage recruitment being confined to the lung tissue but not in the BAL, this is not an unexpected phenotype. Indeed, a recent publication from Poole et al. 2017 addressing the role of B cells in organic dust induced lung inflammation reported similar findings (10). They found no difference in macrophage numbers in the BAL of wild type and B cell-deficient animals after challenge, but in lung tissue after challenge they observed reduced numbers of what they term exudative macrophages (CD11c^{hi} CD11b⁺) in B cell deficient animals compared to wild type. Furthermore, in a study of kidney transplantation in rats, it was observed that B cell deficient animals had reduced intragraft tissue macrophages after allograft transplantation than control recipients (11).

Finally, we would like to highlight the growing body of evidence that iBALT size and number correlates with the severity of COPD in patients. Pertinent work that was published last year analyzing transcriptomics data from COPD patients compared to control smokers revealed that the gene module most associated for COPD was heavily enriched for B cell pathways (12). Additionally, Faner et al. (13) undertook transcriptomics analysis of emphysematous COPD patients versus those only presenting bronchiolitis and clearly demonstrated a distinct B cell signature in the emphysematous patients. Furthermore, recent work by Polverino et al (14) demonstrated that patients with severe COPD (GOLD stages III/IV, mostly with emphysema) had more numerous and larger iBALT structures than more mild patients (GOLD stages I/II). Finally, a publication released this month from the group of James Hogg (15) using a combination of multidetector row computed tomography, micro-computed tomography and histological analysis of lungs from COPD patients, elegantly showed that increased B cell infiltration into the walls of preterminal and pre-preterminal bronchioles of emphysematous COPD patients correlated with destruction of alveolar attachments to these airways.

In summary, we therefore strongly disagree with this reviewer, citing many recent publications with overwhelming evidence supporting our data.

Referee #2 (Comments on Novelty/Model System for Author):

Experiments have been well designed and performed, and address the research hypothesis in a robust manner. Authors have addressed all reviewer concerns appropriately.

We thank the referee most profusely for their positive comments about our manuscript.

References

1. I. D'Hulst A, T. Maes, K. R. Bracke, I. K. Demedts, K. G. Tournoy, G. F. Joos, G. G. Brusselle, Cigarette smoke-induced pulmonary emphysema in scid-mice. Is the acquired immune system required? *Respiratory research* **6**, 147 (2005).
2. F. M. Botelho, G. J. Gaschler, S. Kianpour, C. C. Zavitz, N. J. Trimble, J. K. Nikota, C. M. Bauer, M. R. Stampfli, Innate immune processes are sufficient for driving cigarette smoke-induced inflammation in mice. *American journal of respiratory cell and molecular biology* **42**, 394-403 (2010).
3. K. R. Bracke, F. M. Verhamme, L. J. Seys, C. Bantsimba-Malanda, D. M. Cunoosamy, R. Herbst, H. Hammad, B. N. Lambrecht, G. F. Joos, G. G. Brusselle, Role of CXCL13 in cigarette smoke-induced lymphoid follicle formation and chronic obstructive pulmonary disease. *American journal of respiratory and critical care medicine* **188**, 343-355 (2013).
4. L. J. Seys, F. M. Verhamme, A. Schinwald, H. Hammad, D. M. Cunoosamy, C. Bantsimba-Malanda, A. Sabirsh, E. McCall, L. Flavell, R. Herbst, S. Provoost, B. N. Lambrecht, G. F. Joos, G. G. Brusselle, K. R. Bracke, Role of B Cell-Activating Factor in Chronic Obstructive Pulmonary Disease. *American journal of respiratory and critical care medicine* **192**, 706-718 (2015).
5. R. D. Hautamaki, D. K. Kobayashi, R. M. Senior, S. D. Shapiro, Requirement for macrophage elastase for cigarette smoke-induced emphysema in mice. *Science* **277**, 2002-2004 (1997).
6. S. D. Shapiro, N. M. Goldstein, A. M. Houghton, D. K. Kobayashi, D. Kelley, A. Belaouaj, Neutrophil elastase contributes to cigarette smoke-induced emphysema in mice. *The American journal of pathology* **163**, 2329-2335 (2003).
7. G. John-Schuster, K. Hager, T. M. Conlon, M. Irmeler, J. Beckers, O. Eickelberg, A. O. Yildirim, Cigarette smoke-induced iBALT mediates macrophage activation in a B cell-dependent manner in COPD. *American journal of physiology. Lung cellular and molecular physiology* **307**, L692-706 (2014).
8. E. A. Moseman, M. Iannacone, L. Bosurgi, E. Tonti, N. Chevrier, A. Tumanov, Y. X. Fu, N. Hacohen, U. H. von Andrian, B cell maintenance of subcapsular sinus macrophages protects against a fatal viral infection independent of adaptive immunity. *Immunity* **36**, 415-426 (2012).
9. Y. Zouggari, H. Ait-Oufella, P. Bonnin, T. Simon, A. P. Sage, C. Guerin, J. Vilar, G. Caligiuri, D. Tsiantoulas, L. Laurans, E. Dumeau, S. Kotti, P. Bruneval, I. F. Charo, C. J. Binder, N. Danchin, A. Tedgui, T. F. Tedder, J. S. Silvestre, Z. Mallat, B lymphocytes trigger monocyte mobilization and impair heart function after acute myocardial infarction. *Nature medicine* **19**, 1273-1280 (2013).
10. J. A. Poole, T. R. Mikuls, M. J. Duryee, K. J. Warren, T. A. Wyatt, A. J. Nelson, D. J. Romberger, W. W. West, G. M. Thiele, A role for B cells in organic dust induced lung inflammation. *Respiratory research* **18**, 214 (2017).
11. S. E. Panzer, N. A. Wilson, B. M. Verhoven, D. Xiang, C. D. Rubinstein, R. R. Redfield, W. Zhong, S. R. Reese, Complete B Cell Deficiency Reduces Allograft Inflammation and Intragraft Macrophages a Rat Kidney Transplant Model. *Transplantation*, (2017).
12. J. D. Morrow, X. Zhou, T. Lao, Z. Jiang, D. L. DeMeo, M. H. Cho, W. Qiu, S. Cloonan, V. Pinto-Plata, B. Celli, N. Marchetti, G. J. Criner, R. Bueno, G. R. Washko, K. Glass, J. Quackenbush, A. M. Choi, E. K. Silverman, C. P. Hersh, Functional interactors of three genome-wide association study genes are differentially expressed in severe chronic obstructive pulmonary disease lung tissue. *Scientific reports* **7**, 44232 (2017).
13. R. Faner, T. Cruz, T. Casserras, A. Lopez-Giraldo, G. Noell, I. Coca, R. Tal-Singer, B. Miller, R. Rodriguez-Roisin, A. Spira, S. G. Kalko, A. Agusti, Network Analysis of Lung Transcriptomics Reveals a Distinct B-Cell Signature in Emphysema. *American journal of respiratory and critical care medicine* **193**, 1242-1253 (2016).
14. F. Polverino, B. G. Cosio, J. Pons, M. Laucho-Contreras, P. Tejera, A. Iglesias, A. Rios, A. Jahn, J. Sauleda, M. Divo, V. Pinto-Plata, L. Sholl, I. O. Rosas, A. Agusti, B. R. Celli, C. A. Owen, B Cell-Activating Factor. An Orchestrator of Lymphoid Follicles in Severe Chronic Obstructive Pulmonary Disease. *American journal of respiratory and critical care medicine* **192**, 695-705 (2015).
15. N. Tanabe, D. M. Vasilescu, M. Kirby, H. O. Coxson, S. E. Verleden, B. M. Vanaudenaerde, D. Kinose, Y. Nakano, P. D. Pare, J. C. Hogg, Analysis of airway

pathology in COPD using a combination of computed tomography, micro-computed tomography and histology. *The European respiratory journal* **51**, (2018).

3rd Editorial Decision

26 February 2018

Thank you for the submission of your revised manuscript to EMBO Molecular Medicine. We have evaluated your responses to the referee's last concerns and I am pleased to inform you that we will be able to accept your manuscript pending minor editorial changes.

3rd Revision - authors' response

8 March 2018

Authors made the requested editorial changes.

Corresponding Author Name: Ali Önder Yildirim

Manuscript Number: EMM-2017-08349